# Source influence on emission pathways and ambient PM$_{2.5}$ pollution over India (2015-2050)

Chandra Venkataraman[1,3], Michael Brauer[2], Kushal Tibrewal[3], Pankaj Sadavarte[3,4], Qiao Ma[5], Aaron Cohen[6], Sreelekha Chaliyakunnel[7], Joseph Frostad[8], Zbigniew Klimont[9], Randall V. Martin[10], Dylan B. Millet[7], Sajeev Philip[10,11], Katherine Walker[6], Shuxiao Wang[5,12]

[1]Department of Chemical Engineering, Indian Institute of Technology Bombay, Powai, Mumbai, India
[2]School of Population and Public Health, The University of British Columbia, Vancouver, British Columbia V6T1Z3, Canada
[3]Interdisciplinary program in Climate Studies, Indian Institute of Technology Bombay, Powai, Mumbai, India
[4]Institute for Advanced Sustainability Studies (IASS), Berliner Str. 130, 14467 Potsdam, Germany
[5]State Key Joint Laboratory of Environment Simulation and Pollution Control, School of Environment, Tsinghua University, Beijing 100084, China
[6]Health Effects Institute, Boston, MA 02110, USA
[7]Department of Soil, Water, and Climate, University of Minnesota, Minneapolis–Saint Paul, MN 55108, USA
[8]Institute for Health Metrics and Evaluation, University of Washington, Seattle, WA 98195, USA
[9]International Institute for Applied Systems Analysis, Laxenburg, Austria
[10]Department of Physics and Atmospheric Science, Dalhousie University, Halifax, Nova Scotia B3H 4R2, Canada
[11] NASA Ames Research Center, Moffett Field, California, USA
[12]State Environmental Protection Key Laboratory of Sources and Control of Air Pollution Complex, Beijing 100084, China

*Correspondence to:* Chandra Venkataraman (chandra@iitb.ac.in)

**Abstract.** India currently experiences degraded air quality, with future economic development leading to challenges for air quality management. Scenarios of sectoral emissions of fine particulate matter and its precursors were developed and evaluated for 2015-2050, under specific pathways of diffusion of cleaner and more energy efficiency technologies. The impacts of individual source-sectors on PM$_{2.5}$ concentrations were assessed through systematic simulations of spatially and temporally resolved particulate matter concentrations, using the GEOS-Chem model, followed by population-weighted aggregation to national and state levels. We find that PM$_{2.5}$ pollution is a pan-India problem, with a regional character, not limited to urban areas or megacities. Under present day emissions, levels in most states exceeded the national PM$_{2.5}$ annual standard (40 µg/m3). Sources related to human activities were responsible for the largest proportion of the present-day population exposure to PM$_{2.5}$ in India. About 60% of India's mean population-weighted PM$_{2.5}$ concentrations arise from anthropogenic source-sectors, with the balance from "other" sources, windblown dust and extra-regional sources. Leading contributors are residential biomass combustion, power plant and industrial coal combustion and anthropogenic dust (including coal fly-ash, fugitive road dust and trash burning). Transportation, brick production, and distributed diesel were other contributors to PM$_{2.5}$. Future evolution of emissions under regulations set at current levels and promulgated levels, yielded further deterioration in air-quality in 2030 and 2050. Under an ambitious prospective policies scenario, promoting very large shifts away from traditional biomass technologies and coal-based electricity generation, significant reductions in PM$_{2.5}$ levels are achievable in 2030 and 2050. Effective mitigation of future air pollution in India requires adoption of aggressive prospective regulation, currently not

formulated, for a three-pronged switch away from (i) biomass-fuelled traditional technologies, (ii) industrial coal-burning and (iii) open burning of agricultural residues. Future air pollution is dominated by industrial process emissions, reflecting larger expansion in industrial, rather than residential energy demand. However, even under the most active reductions envisioned, the 2050 mean exposure, excluding any impact from windblown mineral dust, is estimated to be nearly three times higher than
the WHO Air Quality Guideline.

## 1. Introduction

India hosts the world's second largest population (UNDP, 2017), but accounts for only 6% of the world's total primary energy use (IEA, 2015). However, India is an emerging economy with significant growth in a multitude of energy-use activities in industry and transport sectors, as well as in residential, agricultural and informal industry sectors (Sadavarte and
Venkataraman, 2014; Pandey et al. 2014). With expansion in power generation (CEA, 2016) and industrial production (Planning Commission, Government of India, 2013), emissions from these sectors were estimated to have increased about two-fold between 1995-2015 (Sadavarte and Venkataraman, 2014). There is a steady demand for motorized vehicles for both personal and public transport, with an increase in ownership of motorized two-wheeler motorcycles and scooters and four-wheeler cars (MoRTH, 2012.), in both rural and urban areas. Traditional technologies, and the use of solid biomass fuels, are
widespread in the residential sector (cooking with biomass fuel cook stoves and lighting with kerosene wick lamps), the agricultural sector (open burning of agricultural residues for field clearing), and the informal industry sector, (brick production, processing of food and agricultural products). Ambient $PM_{2.5}$ (particulate matter in a size fraction with aerodynamic diameter smaller than 2.5 µm) concentrations are influenced by emissions of both primary or directly emitted $PM_{2.5}$, and its precursor gases, including $SO_2$, $NH_3$, $NO_x$, and NMVOCs (Non-methane volatile organic compounds), whose atmospheric reactions
yield secondary particulate sulphate, nitrate and organic carbon, while reactions of $NO_x$ and NMVOCs also increase ozone levels. Ozone precursor gases and particulate black carbon and organic carbon (BC and OC) are identified in the list of short-lived climate pollutants or SLCPs (CCAC, 2014).

Air quality is a public health issue of concern in India. According to the World Health Organization (WHO), 37 cities from
India feature in a global list of 100 world cities with the highest $PM_{10}$ (PM with aerodynamic diameter <10 µm) pollution, with cities like Delhi, Raipur, Gwalior, and Lucknow listed among the world's top 10 polluted cities (WHO, 2014; further details in Fig. S6 of supplement). Recent studies (Ghude et al. 2016; Chakraborty et al. 2015), have built upon products of the Task Force on Hemispheric Transport of Air Pollutants (TF-HTAP), using HTAP emission inventories (for 2010) in a regional chemistry model to address air quality in India. Widespread $PM_{2.5}$ and O3 pollution was found under present-day emission
levels, which considerably impact human mortalities and life expectancy. To extend the understanding of ambient air pollution to multiple (regional and national) scales, for multiple pollutants, methods which combine chemical transport modelling, with data from satellite retrievals combined with available monitoring data, have been developed (van Donkelaar et al., 2010; Brauer

et al. 2012, 2016; Dey et al., 2012; Shaddick et al., 2018) and can be used to evaluate current levels and trends. The latest GBD 2015 estimates indicate that the population-weighted mean $PM_{2.5}$ concentration for India as a whole was 74.3 µg/m$^3$ in 2015, up from about 60 µg/m$^3$ in 1990 (Cohen et al., 2017). At current levels, 99.9% of the Indian population is estimated to live in areas where the World Health Organization (WHO) Air Quality Guideline of 10 µg/m$^3$ was exceeded. Nearly 90% of people lived in areas exceeding the WHO Interim Target 1 of 35 µg/m$^3$.

Strategies for mitigation of air pollution require understanding pollutant emissions, differentiated by emitting sectors and by sub-national regions, representing both present day conditions and future evolution under different pathways of growth and technology change. Future projections of emissions, for climate relevant species, are available in the representative concentration pathway (RCP) scenarios (Fujino et al. 2006; Clarke et al. 2007; Van Vuuren et al. 2007; Riahi et al. 2007; Hijioka et al. 2008), more recently for the Shared Socioeconomic Pathways (SSPs) scenarios (Riahi et al., 2017; Rao et al., 2017), while primary $PM_{2.5}$ is included in inventories like ECLIPSE (Klimont et al., 2017, 2018). Inventories developed for HTAP_v2 (Janssens-Maenhout et al. 2015) address emissions of a suite of pollutants for 2008 and 2010. These scenarios and emission datasets are developed through globally consistent methodologies, leaving room for refinement through more detailed regional studies. Thus, in this work we develop and evaluate sectoral emission scenarios of fine particulate matter and its precursors and constituents from India, during 2015-2050, under specific pathways of diffusion of cleaner and more energy efficiency technologies. The work is broadly related to HTAP scientific questions including understanding of (i) sensitivity of regional $PM_{2.5}$ pollution levels to magnitudes of emissions from source-sectors and (ii) changes in $PM_{2.5}$ levels as a result of expected, as well as ambitious, air pollution and climate change abatement efforts. The impacts of individual source-sectors on $PM_{2.5}$ concentrations is assessed through simulation of spatially and temporally resolved particulate matter concentrations, using the GEOS-Chem chemical transport model, followed by aggregation to population-weighted concentrations (estimated as the sum of product of concentration and population for each grid divided by the total population) at both national and state levels.

Section 2 discusses the development of the emission inventory, disaggregated by sector, for the year 2015 and future projections to 2050; Section 3 describes the GEOS-Chem model, the simulation parameters and evaluation; Section 4 discusses simulated $PM_{2.5}$ concentration by sector, at national and state levels under present day and future emission scenarios; and the last section discusses findings and conclusions.

## 2. Present day and future emissions

### 2.1. Present day emissions (2015)

An emission inventory was developed for India, for the year 2015, based on an "engineering model approach" using technology-linked energy-emissions modelling adapted from previous work (Pandey and Venkataraman 2014; Pandey et al.

2014; Sadavarte and Venkataraman, 2014), to estimate multi-pollutant emissions including those of $SO_2$, $NO_x$, $PM_{2.5}$, black carbon (BC), organic carbon (OC), and non-methane volatile organic compounds (NMVOCs). An engineering model approach, goes beyond fuel divisions and uses technology parameters for process and emissions control technologies, including technology type, efficiency or specific fuel consumption, and technology-linked emission factors (g of pollutant/ kg of fuel)

to estimate emissions.

The inventory disaggregates emissions from technologies and activities, in all major sectors. Plant level data (installed capacity, plant load factor, and annual production) are used for 830 individual large point sources, in heavy industry and power generation sectors, while light industry activity statistics (energy consumption, industrial products, solvent use, etc.) are from

sub-state (or district) level (CEA 2010; CMA 2007a,b, 2012; MoC 2007; FAI 2010; CMIE 2010; MoPNG 2012; MoWR 2007). Technology-linked emission factors and current levels of deployment of air pollution control technologies are used. Vehicular emissions include consideration of vehicle technologies, vehicle age distributions, and super-emitters among on-road vehicles (Pandey and Venkataraman, 2014). Residential sector activities comprise of cooking and water heating, largely with traditional biomass stoves; lighting, using kerosene lamps; and warming of homes and humans, with biomass fuels. Seasonality included

for water heating and home warming. The "informal industries" sector includes brick production (in traditional kiln technologies like the Bull's trench kilns and clamp kilns, using both coal and biomass fuels) and food and agricultural product processing operations (like drying and cooking operations related to sugarcane juice, milk, food-grain, jute, silk, tea, and coffee). In addition, monthly mean data on agricultural residue burning in fields, a spatio-temporally discontinuous source of significant emissions, were calculated using a bottom-up methodology (Pandey et al. 2014). Spatial proxies used to estimate

gridded emissions over India are described in Table S1 of the supplement.

India emissions for 2015 of $PM_{2.5}$, BC, OC, $SO_2$, $NO_x$, and NMVOCs by sector (Fig. 1) arose from three main sources: (i) residential biomass fuel use (for cooking and heating); (ii) coal burning in power generation and heavy industry; and (iii) open burning of agricultural residues for field clearing. Table 1 provides a description of sectors and constituent source

categories. Emissions linked to incomplete fuel combustion, including $PM_{2.5}$ (9.1 Mt/yr, or million tonnes per year), BC (1.3 Mt/yr) and OC (2.3 Mt/y) and NMVOCs (33.4 Mt/yr), arose primarily from traditional biomass technologies in the residential sector (for cooking and heating), the informal industry sector (for brick production and for food and agricultural produce processes), as well as from agricultural residue burning. Emissions of $SO_2$ (8.1 Mt/yr) and $NO_x$ (9.5 Mt/yr) arose largely from coal boilers in industry and power sectors and from vehicles in the transport sector. Emissions of CO are included in the

inventory (Pandey et al., 2014; Sadavarte et al., 2014), however, CO was not input to the GEOS-Chem simulations, since it is not central to atmospheric chemistry of secondary $PM_{2.5}$ formation on annual time-scales.

Detailed tabulations of 2015 emissions of each pollutant at the state level are provided in Table S2 of the supplement. Uncertainties in the activity rates, calculated analytically using methods described more fully in previous publications (Pandey and Venkataraman 2014; Pandey et al. 2014; Sadavarte and Venkataraman, 2014) are shown in Table S3 of the supplement.

## 2.2. Future emission pathways (2015-2050)

### 2.2.1. Description of future emission scenarios

We develop and evaluate three future scenarios which extend from 2015-2050, which are likely to bound the possible amplitude of future emissions, based on the expected future evolution of sectoral demand, following typical methods in previous studies (Cofala et al., 2007; Ohara et al., 2007). These include a reference (REF) scenario and two scenarios (S2 and S3) representing different levels of deployment of high-efficiency, low-emissions technologies (Table 2). The scenarios capture varying levels

of emission control, with no change in current (2015) regulations, corresponding to very slow uptake of new technology (REF), adoption of promulgated regulations, corresponding to effective achievement of targets (S2), and adoption of ambitious prospective regulations, corresponding to those well beyond promulgated regulations (S3). In both S2 and S3, despite expanding sectoral demand, there is reduced energy consumption from adoption of clean energy technologies, at different levels.

The methodology for emission projection includes estimation of future evolution in (i) sectoral demand, (ii) technology mix, (iii) energy consumption, and (iv) technology-linked emission factors (Fig. S1 of supplement). Activity levels in future years by source category (e.g. GWh installed capacity in power, vehicle-km travelled in transport, industrial production, e.g. in tons, population of users in residential), were apportioned to various technology divisions, using assumed evolving technology mix,

for three different scenarios. Activity at the technology division level was used to derive corresponding future energy (and fuel) consumption and related emissions using technology-based emission factors.

With 2015 as the base year, growth rates in sectoral demand were identified for thermal power plants, industries, residential, brick kilns and informal industries, on-road transportation and agricultural sectors for 2015-2030 and 2030-2050 (Table S4 of

supplement). Sectoral growth, estimated as ratios of 2050 to 2015 demand, were 5.1, 3.8, 3.2, 1.3, 1.4 respectively, for building sector, electricity generation, heavy industries, residential sector, and agricultural residue burning, with the largest growth in the building and electricity generation sectors (Fig. S2 of supplement).

Table 2 shows regulation levels for different sectors under the three scenarios, through to 2050. The REF and S2 scenarios

capture both energy efficiency and emissions control, continuing under current regulation, or broadly under promulgated future policies. The S2 scenario assumes shifts to non-fossil generation which would occur under India Nationally Determined Contribution (India's NDC, 2015) in the power sector, consistent with a shift to 40% renewables including solar, wind and

hydro power by 2030 (NDC, 2015). The NDC goals of India are suggested to be realistic (CAT, 2017; Ross and Gerholdt, 2017), with achievement of non-fossil share of power generation projected to lie between 38%-48% by 2030, as well as adoption of tighter emission standards for desulphurization and de-NOx technologies in thermal plants (MoEFCC, 2015), at a rate consistent with expected barriers (CSE, 2016). Further, changes assumed in the transport sector reflect promulgated growth in public vehicle share (NTDPC, 2013; Guttikunda and Mohan, 2014; NITI Aayog, 2015) and promulgated regulation (Auto Fuel Policy Vision 2025, 2014, MoRTH, 2016), along with realistic assumptions of implementation lags in adoption of BS VI standards (ICRA 2016). Other assumptions include modest increases in industrial energy efficiency under the perform achieve and trade (PAT) scheme (Level 2, IESS, Niti Aayog, 2015 ); modest increases in non-fired-brick walling materials (UNDP, 2009; Maithel, personal communication, 2016); slow shift to more efficient residential energy technologies and fuels (Level 2, IESS, Niti Aayog, 2015); and minor reduction in agricultural residue burning.

However, in the S3 scenario, adoption of ambitious regulation, well beyond those currently promulgated is assumed. This includes very significant shifts to non-fossil power generation (Anandarajah and Gambhir 2014; Shukla and Chaturvedi 2012; Level 4, IESS, Niti Aayog, 2015); near-complete shift to high efficiency industrial technologies (MoP 2012, Level 4, IESS, Niti Aayog, 2015); large public vehicle share (NITI Aayog, 2015), energy efficiency improvements in engine technology (MoP, 2015), large share of electric and CNG vehicles (NITI Aayog, 2015); complete switch to LPG/PNG or biogas or high-efficiency gasifier stoves for residential cooking and heating (Level 4, IESS, Niti Aayog, 2015) and to solar and electric lighting (National Solar Mission, 2010) by 2030; significant (by 2030) and complete (by 2050) phase-out of agricultural residue burning, through a switch to mulching practices (Gupta, 2014). Further details of the shift in technologies can be found in Table S5 of supplement and related discussion in supplementary information (see supplement, section S2.3).

As alluded to earlier, there is a reduction in total energy consumption in future years, despite increase in activity, in scenarios S2 and S3, which assume large deployment of high-efficiency energy technologies. The projected energy demand under the three scenarios (Fig. S3, supplement section S2.4) is in general agreement with published work (Anandarajah and Gambhir 2014; Chaturvedi and Shukla 2014; Parikh 2012; Shukla et al. 2009), of 95 EJ to 110 EJ for reference scenarios (Parikh, 2012; Shukla and Chaturvedi 2012) and 45-55 EJ for low carbon pathways (Anandarajah and Gambhir 2014; Chaturvedi and Shukla 2014) in 2050. Projections of $CO_2$ emissions to 2050, of 7200 Mt $yr^{-1}$ in REF and 2000 Mt $yr^{-1}$ in S3, are broadly consistent with published 2050 values of 7200-7800 million tonnes $y^{-1}$ $CO_2$ for reference cases, and 2500-3400 million tonnes $y^{-1}$ $CO_2$ under different low carbon scenarios (Anandarajah and Gambhir 2014; Shukla et al. 2009).

Technology based emission factors, for over 75 technology/activity divisions, are described in previous publications (Pandey et al. 2014; Sadavarte and Venkataraman 2014). In addition to fuel combustion, emissions are estimated from industrial "process" activities predominant in industries such as those producing cement and non-ferrous metals, and refineries producing iron and steel (Table S8, supplement section S2.5). In fired-brick production, recently measured emission factors for this sector

of $PM_{2.5}$, BC and OC (Weyant et al.,2014) are used (Table S8 of supplement), while for gases, in the absence of measurements from brick kilns, those of coal stokers are used. In the transport sector, emission factors for seven categories of vehicles, across two vintage classes, were applied to a modelled on-road vehicle age distribution (Pandey and Venkataraman, 2014). For future emissions, recommendations from the Auto Fuel Policy 2025 (Auto Fuel Vision and Policy 2025) along with accounting of

the measures to leapfrog directly to BS-VI for all on-road vehicle categories (MoRTH, 2016). To be consistent with our scenario descriptions, the REF scenario still takes into account the BS-V standards for 2030 and 2050 while the effect of dynamic policy reforms is reflected in the tech-mix in S2 and S3 scenarios by assuming different levels of BS-VI. The  share of BS-VI is kept at modest levels owing to delay in availability of BS-VI compliant fuels and difficulties in making the technologies adaptive to Indian road conditions as well as cost-effective (ICRA, 2016), however, would not affect emission

factors significantly (Table S8 of supplement).

### 2.2.2. Estimated emission evolution (2015-2050)

The net effect of scenario based assumptions is that under the REF scenario, emissions are projected to increase steadily over time. Under the S2 scenario, they are also projected to increase but at a slower rate. Only under the most ambitious scenario, S3, are appreciable reductions in emissions of the various air pollutants expected.

Emissions of $PM_{2.5}$ evolve from present-day levels of 9.1 Mt/yr to 2050 levels of 18.5, 11.5 and 3.0 Mt/yr, respectively, in the three scenarios (Fig. 2 a, b, c). These arise from three main sources: (i) traditional biomass technologies in residential, brick production and informal industry, (ii) coal burning in power generation and heavy industry, and (iii) open burning of agricultural residues for field clearing. In Fig. 1-3, emissions shown are only from agricultural burning, while those from forest

and wildfires, taken from global products, described later, are input to the simulations. In all future scenarios, there is faster growth of industry and electricity generation than of residential energy demand; the former which contribute nearly 60–70% of future emissions. Thus, controlling emissions of $PM_{2.5}$ should come from these sectors. As is quite evident (Fig. 2 b and c), assuming large shifts to non-coal power generations in scenarios S2 (40-60%) and S3 (75-80%) in S3 contribute most to reductions in future emissions of $PM_{2.5}$. Further reductions in emissions are obtained through shifts to cleaner technology and

fuels in the residential sector such as use of gasifiers and LPG for cooking, electricity and solar devices for lighting and heating, and complete phase out of open burning of agricultural waste. Black carbon and co-emitted organic carbon have very similar sources with the largest emissions arising from traditional biomass technologies in the residential and informal industry sectors and from agricultural field burning. Future reductions in BC (Fig. 2d, e, f) and OC (Fig. 2g, h, i) emissions result from a number of policies addressing residential and informal industry sectors as well as agricultural practices. These includes actions

that enable a shift to cleaner residential energy solutions and a shift away from fired-brick walling materials toward greater use of clean brick production technologies, as well as a shift away from agricultural field burning through the introduction of mulching practices (assumed in S3). Future increases in transport demand could lead to increased BC emissions from diesel-powered transport, thus providing an important decision lever in favour of the introduction of compressed natural gas (CNG)

or non-fossil-electricity powered public transport (in S3). While diesel particle filters provide a technology for diesel PM and BC control, challenges remain including the supply of low-sulphur fuel and compliance with NOx emission standards.

Emissions of SO2 increase in 2050 (Fig. 3d, e, f) to 41.4-20.7 Mt/yr under REF and S2, but stabilize at 7.5 Mt/yr under S3. Under both REF and S2 scenarios (Fig. 3a, b, c), emission growth of SO2 is driven by growth in electricity demand and industrial production, while reduction is driven by a shift to non-carbon power generation (nuclear, hydro, solar, and wind) and modest adoption of flue gas desulphurization technology. In December 2015, the Indian Ministry of Environment and Forests issued new norms for thermal plants with emission standards for SO2 and NOx (MoEFCC, 2015). Our assumption here of negligible flue gas desulphurization technology follow from reported barriers to adoption of desulphurization and de-NOx technologies (CSE, 2016). Little progress was found (CSE, 2016) in the implementation of new standards, from lack of technology installation/operation information, space for retrofitting and clarity on cost recovery. Transport-related SO2 emissions are negligible in all scenarios. Emissions of NOx increase in 2050 (Fig. 3d, e, f) to 31.7-18.4 Mt/yr under REF and S2, but stabilize at 10.5 Mt/yr under S3. The emissions shares are dominated by thermal power and the transport sector, and grow with sectoral growth under the first two scenarios. Under future scenarios, the demand in passenger-km increases twice that in ton-km of freight, thus leading in 2050 to significantly greater passenger (7000-10000 billion passenger-km, in different scenarios), than freight (2300-2800 billion ton-km) transport provided by diesel. This makes shifts away from diesel based public transport important. Thus, under the S3 scenario, shifts in the transport sector to tighter emission standards for vehicles and a greater share of CNG in public transport, as well as, in the power sector, to non-fossil power generation, reduce NOx emissions. Owing to the large shift away from fossil-power, the use of selective catalytic reduction (SCR) technology for NOx control is not considered. A non-negligible, approximately 20%, share is from residential, agricultural field burning and brick production sectors, which is reduced in magnitude by the adoption of mitigation based largely on cleaner combustion technologies. Emissions of NMVOCs increase in 2050 to 16.3 Mt/yr under the REF scenario, but decrease to about 3.8 Mt/yr under S3 (Fig. 3g, h, i). In the S3 scenario, mitigation in residential, transport and open burning emissions offsets more than two-thirds of present-day NMVOC emissions. Industrial emissions of NMVOC, arising primarily from solvent use, are almost constant at 2 Mt/yr across scenarios, providing further potential for mitigation. However, a shift to public transport based on heavy-duty CNG vehicles drives the increase in NMVOC emissions from the transport sector, from their significantly larger emissions factors, compared to those of heavy duty diesel. Therefore, alternate modes and technologies in the transport sector need further attention.

Anthropogenic dust (Philip et al. 2017), defined here as mineral constituents of pollution particles, including coal fly-ash and mineral matter in trash burning and biomass burning emissions, contributes about 30% of Indian $PM_{2.5}$ emissions in the base year 2015 i.e. about ~3 Mt/yr. In future scenarios REF and S2, respectively, anthropogenic dust contributes 6.0 and 4.6 Mt/yr in 2030 and 12.0 and 6.8 Mt/yr in 2050, arising primarily (60–85%) from coal fly-ash, with the balance from fugitive on-road dust and waste burning. In the highest-control S3 scenario, anthropogenic dust emissions were reduced to about 1.8 Mt/yr, in

both 2030 and 2050. This results from the assumed significant shift to 80–85% non-coal thermal power generation, leading to large reductions in coal fly-ash emissions. Thus, in the S3 scenario anthropogenic dust emissions arise largely from on-road fugitive dust and waste burning (over 50%), with a lower contribution from coal fly-ash (35-40%).

5   Emission datasets for India in global emission inventories have been developed either through combination of regional inventories for specific base years (Janssens-Maenhout et al., 2015) or using integrated assessment models, e.g., the GAINS model (Amann et al., 2011), to generate scenarios of air pollutants (Klimont et al., 2009, 2017, 2018; Purohit et al., 2010; Stohl et al., 2015). Indian emissions for 2008 and 2010 under the HTAP_v2 framework (Janssens-Maenhout et al., 2015), originate from the MIX inventory (Li et al., 2017), based on earlier Asia inventories like INTEX-B (Lu et al., 2011; Lu and Streets, 2012) and REAS (Kurokawa et al., 2013). Inconsistencies are reported from merging datasets, calculating different pollutants using differing assumptions (Li et al., 2017). The datasets do not include some important regional emission sources like the open burning of agricultural residues (Janssens-Maenhout et al., 2015). Recent global emissions from ECLIPSE V5 (Stohl et al., 2015; http://www.iiasa.ac.at/web/home/research/researchPrograms/air/ECLIPSEv5.html), driven by HTAP objectives to improve representation of aerosols emissions in IAMs (Keating, 2015), were reported to have problems over India including underestimation of BC and trace gas magnitudes and inaccuracies in spatial distribution (Stohl et al., 2015). The present dataset overcomes some of these limitations, using consistent assumptions to calculate a number of pollutants, including all sectors in global inventories, as well as, agricultural residue burning emissions, industrial process emissions, while providing for finer spatial resolution using district level data and more relevant spatial proxies. Emission magnitudes of $PM_{2.5}$ and precursors in present inventory are in good agreement with those in ECLIPSE V5a for 2010, however, those of precursor gases are somewhat lower (about 30%) than those in HTAP_v2 (2010) and REAS 2.1 (2008) (Section 2.6 of supplement).

Future emissions of particulate matter ($PM_{2.5}$ and constituents, BC and OC) and precursor gases (SO2, NOx and NMVOC) estimated here were compared with the more recent sets of scenarios developed with the GAINS model in projects addressing global air pollution trajectories until 2050, i.e., the 'Current Legislation scenario' (CLE) of ECLIPSE V5a (Klimont et al., 2017, 2018;) and the 'New Policies Scenario' (NPS) of World Energy Outlook (IEA, 2016). These scenarios rely on different energy projections; Energy Technology Perspective study (IEA, 2012) was used in ECLIPSE V5a and World Energy Outlook 2016 in the IEA study. Furthermore, the assumptions about air pollution legislation vary with IEA study considering within the 'New Policies Scenario' recently adopted, announced or intended policies, even where implementation measures are yet to be fully defined. In general, lower emissions in GAINS-WEO2016-NPS (IEA, 2016) are attributed to the successful implementation of new emission regulations in power and transport sectors, decreased use of biomass fuel in residential sector and phase-out of kerosene lamps. We compare S2 and S3 scenarios in the present study to the baseline scenarios from the above studies (shown in Fig. 2 and Fig. 3).

For SO2 and NOx, emission trajectories in the S2 scenario are similar to those in ECLIPSE V5a - CLE, while emissions in the S3 scenario resemble those in GAINS-WEO2016 –NPS where newly proposed SO2 and NOx regulations for thermal power plants and implementation of BS-VI in transportation is included. In fact, also the absolute level of emissions estimated for 2015 is comparable to this study (Fig. 3a, d); though GAINS estimates are slightly higher for SO2 and lower for NOx owing

primarily to differences in emission factors for coal power plants. Bottom-up estimates of SO2 emissions from our inventory (Pandey et al., 2014; Sadavarte et al., 2014) are consistent with the recent estimates from the satellite based study (Li et al., 2017) from 2005-2016, both showing a steady growth. Present day emissions of SO2 (8.1 Mt yr-1) are at the lower end of the range of 8.5-11.3 Mt yr-1suggested by Li et al. 2017. Large future increases in SO2 emissions, estimated here in the REF and S2 scenarios are consistent with findings of Li et al. 2017.

For particulate matter species, the GAINS model estimates lower 2015 emissions mostly because of the differences for residential use of biomass as well as emissions from open burning. However, considering the uncertainties associated with quantification of biomass use and emission factors (e.g., Bond et al., 2004; Klimont et al., 2009, 2017; Venkataraman et al., 2010) the differences are acceptable. The future evolution of emissions of BC and OC shows similar features among the studies

with S2 comparable to ECLIPSE V5a-CLE and S3 to GAINS-WEO2016-NPS, however the S3 scenario brings much stronger reduction due to faster phase-out of kerosene for lighting and stronger reduction of biomass used for cooking; the latter feature is especially visible for emissions of OC (Fig. 2d, g). For total $PM_{2.5}$ (Fig. 2a) scenarios developed with the GAINS model do not show a very large difference and fall short of the reductions achieved in the S3 case where significant mitigation reduction is not achieved in residential sector for also in power sector and industry which in GAINS are either already controlled in the

baseline (power sector) or continue to grow, industrial processes offsetting the benefits of reduction in other sectors.

Emissions of NMVOCs (Fig. 3g) monotonically increase in ECLIPSE V5a-CLE, becoming higher than those in S2, by 2030, which however, mimic those in GAINS-WEO2016-NPS, through to 2050. While there is also a fairly large difference in estimate for the base year (mostly due to residential combustion of biomass, open burning, and solvent use sector), obviously

the assumptions about the future policies are different as both ECLIPSE V5a and IEA study include more conservative assumptions about reduction of biomass use and eradication of open burning practices while at the same time continued growth in industrial emissions, i.e., solvent applications. Further analysis of differences between the S2 scenario and the ECLIPSE V5a-CLE and GAINS-WEO2016-NPS is shown in the supplement (Fig. S5).

Further, the emission projections were also compared with emissions estimated in the four representative concentration pathways (RCP) scenarios adopted by the IPCC as a common basis for modelling future climate change (Fujino et al. 2006; Clarke et al. 2007; Van Vuuren et al. 2007; Riahi et al. 2007; Hijioka et al. 2008). The RCP scenarios were designed to represent a range of possible future climate outcomes in terms of radiative forcing watts per square meter (Wm-2) values (2.6, 4.5, 6.0, and 8.5) in 2100 relative to pre-industrial levels.  Overall, Indian emissions of SO2, NOx, and BC estimated here in

the REF and S2 scenarios, which do not apply stringent controls, were 2 to 3 times higher than the largest emissions estimated in the RCP8.5 scenario in 2030 and 2050, as a result of differences in assumptions made or in the list of sources included (Table S9 of supplement). As all RCP scenarios considered principally one type of air pollution trajectory assuming that air pollutant emissions will be successfully reduced with economic growth. Consequently, in the longer term the range of

outcomes is fairly similar among RCPs (Amann et al., 2013; Rao et al. 2017). Emissions of these species in the S3 scenario, with the most stringent controls, were in agreement with either RCP8.5 or RCP 4.5 scenario emissions. Emissions of OC in the REF and S2 scenarios and of NMVOCs in the S2 and S3 scenarios were in agreement with the ranges estimated in the RCP4.5 and RCP8.5 scenarios. Emissions of SO2 estimated here for the highest-control scenario, S3, agreed with those from RCP 4.5 in 2030 and RCP 8.5 in 2050, due to similar assumptions of over 80% non-coal electricity generation. However, the

S2 and REF scenarios estimated much larger emissions. Further details are presented in section S2.6 of supplement.

### 3. Model simulations and evaluation

The emissions were used with GEOS-Chem model (www.geos-chem.org) to calculate pollutant concentration fields in space and time. The GEOS-Chem model has been previously applied to study $PM_{2.5}$ over India (e.g., Boys et al. 2014; Kharol et al. 2013; Philip et al. 2014a; Li et al. 2017) including relating satellite observations of aerosol optical depth to ground-level $PM_{2.5}$

for the GBD assessment (Brauer et al. 2012, 2016; van Donkelaar et al. 2010, 2015, 2016). The simulations undertaken in this work represent one of the finest resolution efforts to date to both represent India, and global scale processes.

In addition to the emissions described in section 2.2.2, other emissions such as open burning except agricultural residue burning, which includes forest fires were derived from the global GFED-4s database (Akagi et al. 2011; Andreae et al. 2001;

Giglio et al. 2013; Randerson et al. 2012; van der Werf et al. 2010). In addition to the species in this inventory, ammonia or $NH_3$ emissions, important for calculating secondary particulate matter, were taken from the MIX emission inventory (Li et al. 2017; http://meicmodel.org/dataset-mix.html). Emissions of NH3 arise primarily from sources like animal husbandry, not addressed in the present inventory. Therefore, they are taken from (Li et al., 2017). Owing to large uncertainties in future emissions, these were held the same in future scenarios, as for 2015. Emission magnitudes of NH3 could affect secondary

nitrate, which typically contributes to less than 5% of $PM_{2.5}$ mass (Fig. 5a, c; Kumar and Sunder Raman, 2016; Ram and Sarin, 2011; Rastogi et al., 2016), thus not influencing overall results in any significant manner. The model solves for the temporal and spatial evolution of aerosols and gaseous compounds using meteorological data sets, emission inventories, and equations that represent the physics and chemistry of the atmosphere. Version 10.01 is used here. Total NMVOC emissions from India were taken from Sarkar et al (2016). The GEOS-Chem model speciation (Table S10, supplementary material), into eight

species, was applied for further input to the photochemical module. The simulation of $PM_{2.5}$ includes the sulphate–nitrate–ammonium–water system (Park et al. 2004), primary (Park et al. 2003) and secondary (Henze et al. 2006, 2008; Liao et al. 2007; Pye et al. 2010) carbonaceous aerosols, mineral dust (Fairlie et al. 2007), and sea salt (Alexander et al. 2005). The

GEOS-Chem model has fully coupled ozone–$NO_x$–hydrocarbon chemistry and aerosols including sulphate ($SO_4^{2-}$), nitrate ($NO_3^-$), ammonium ($NH_4^+$) (Park et al. 2004; Pye et al. 2009), organic carbon (OC) and black carbon (BC) (Park et al. 2003), sea salt (Alexander et al. 2005), and mineral dust (Fairlie et al. 2007). For these simulations we also included the $SO_4^{2-}$ module introduced by Wang and colleagues (2014). Partitioning of nitric acid ($HNO_3$) and ammonia between the gas and aerosol

phases is calculated by ISORROPIA II (Fountoukis and Nenes 2007). Secondary organic aerosol formation includes the oxidation of isoprene (Henze and Seinfeld 2006), monoterpenes and other reactive volatile organic compounds (Liao et al. 2007), and aromatics (Henze et al. 2008).

The South Asia nested version of GEOS-Chem used here was developed by Sreelekha Chaliyakunnel and Dylan Millet (both

of the University of Minnesota) to cover the area from 55°E to 105°E and from 0°S to 40°N, and to resolve the domain of South Asia at a resolution of 0.5° × 0.67° (approximately 56 × 74 km at equator) with dynamic boundary conditions using meteorological fields from the NASA Goddard Earth Observation System (GEOS-5). The boundary fields are provided by the global GEOS-Chem simulation with a resolution of 4° latitude and 5° longitude (approximately 445 × 553 km at equator), which are updated every three hours. We have corrected the too-shallow nighttime mixing depths and overproduction of $HNO_3$

in the model following Heald and colleagues (2012) and Walker and colleagues (2012). We applied the organic mass to organic carbon ratio in accordance with findings from Philip et al. (2014b). A relative humidity of 50% was used to represent simulated $PM_{2.5}$ measurements in India. South Asia nested meteorological fields were not yet available post-2012 due to a change in the GEOS assimilation system in 2013. Therefore, we conducted standard simulations to test meteorology from the years 2010 to 2012. We chose the year 2012 as our meteorology year, as the simulation results using this year best represented the mean

$PM_{2.5}$ concentration from 2010 to 2012. A three month initialization period was used to remove the effects of initial conditions.

To estimate the impacts of individual sources, simulations were made using total emissions from all sources, along with sensitivity simulations (Table 1) for major sources. Sources included in the standard simulation, however, not separately addressed in sensitivity simulations, termed "other" include residential lighting with traditional kerosene lamps and informal

industry (food and agro-product processing). Primary particulate matter is largely composed of carbonaceous constituents (black carbon and organic matter) and mineral matter. Mineral matter from combustion and industry are calculated as the difference between emitted $PM_{2.5}$ mass and the sum of black carbon and organic matter, each calculated from respective emission factors and lumped along with urban fugitive dust, evaluated in a previous study (Philip et al. 2017), are termed anthropogenic fugitive dust or ADST. For sensitivity simulations, the total coal-related emissions, industrial coal-related

emissions, and emissions from other major sectors are removed respectively from the inventory in each scenario. The global and nested grid models of GEOS-Chem were then run in sequence using the new inventories. These sensitivity simulation results therefore depict the ambient $PM_{2.5}$ concentrations with each emission sector shut off. The differences of the standard and sensitivity simulations were analyzed to produce contributions of the individual sectors to ambient $PM_{2.5}$ concentrations. By comparing the difference in simulated ambient concentrations between the standard and sensitivity simulations, we

therefore consider in our analyses the complex nonlinear relationships between emissions and ambient concentrations and the nonlinear atmospheric chemistry affecting particle formation.

The GEOS-Chem simulations made here include those for primary aerosol emissions; secondary sulphate, nitrate, and
ammonium; and secondary organic aerosol, going beyond previous simulations made on regional scales over India (e.g., Sadavarte et al. 2016), which were limited to secondary sulphate and a smaller list of sources in the emissions inventory, addressing only a few months in the year. Model predicted concentrations of $PM_{2.5}$ (Fig. 4) and its chemical constituents (Fig. 5) were evaluated against available $PM_{2.5}$ measurements, satellite observations of columnar aerosol optical depth (AOD), and available monthly chemical composition measurements (Kumar and Sunder Raman 2016; Ramachandran and Kedia 2010;
Ramachandran and Rajesh 2007). Model performance was evaluated through normalized mean bias (NMB) (Eq. 1) for pairs of model predicted concentrations (M) and corresponding observed concentrations (O), at given locations and for the same averaging period:

$$\text{Normalized Mean Bias} = \frac{\sum_1^n (M - O)}{\sum_1^n (O)} \tag{1}$$

The evaluation of the seasonal cycle of simulated $PM_{2.5}$ is inhibited by the paucity of measurements. Evaluation of the $PM_{2.5}$
seasonal variation reveals an overall general consistency between the simulation and observations. However, some of the largest concentrations, e.g. at Delhi (28.6° N, 77.1° E) and Kanpur (26.4° N, 80.3° E), were somewhat underestimated. The model captures AOD distribution over large parts of India, compared to measurements from MODIS (Fig. 4b; NMB of -33%) but appears to have an underestimation in the northwest, implying underestimation in modelled windblown dust emissions in the Thar desert. However, the evaluation may be interpreted with caution, from differences arising from sensor (e.g. MODIS
and MISR) variability in the AOD product both spatially and temporally over India (Baraskar et al., 2016), as well as, lack of coincident sampling of model with satellite observations.

Evaluation was also explored against monthly mean chemical composition measurements (Fig. 5) at a regional background site (Bhopal, 23.2° N, 77.4° E; Fig. 5a, b, c; $PM_{2.5}$, sulphate, nitrate; methods described in Kumar and Sunder Raman, 2016)
and a western urban site (Ahmedabad, 23.0° N, 72.5° E; Fig. 5d, BC; aethalometer measurements in Ramachandran and Rajesh, 2007). The simulation captures monthly $PM_{2.5}$ and species mean concentrations satisfactorily during non-winter months at the two sites, but with some underestimation in the winter months. While sensitivity simulations for nitrate (not shown) increased nitrate concentrations in north India, they were largely unchanged in central India, evident in the underestimation of nitrate (NMB = -68%) at Bhopal. The spatial distribution of particulate species (not shown) reflects the interplay of emission density
distributions with transport processes, with sulphate showing a predominance in central India and to the east where there is a prevalence of thermal power generation, but BC and organic matter showing a predominance in northern India, where there is a prevalence of traditional biomass fuelled residential energy technologies. The findings here are broadly consistent with earlier work (Sadavarte et al. 2016) which showed large surface concentrations of sulphate, organic carbon and dust over north India.

As discussed earlier, NMVOC emissions from India were taken from a recent technology-linked inventory, deployed in WRF-CAMx and evaluated with satellite and in-situ observations (Sarkar et al. 2016). However, uncertainties still remain to be addressed in the calculation of secondary $PM_{2.5}$ constituents, especially secondary organic aerosols, whose precursor NMVOC

emissions in developing countries, are still uncertain from lack of speciation measurements under combustion conditions (Roden et al., 2006; Martinsson et al., 2015) typically encountered in traditional technologies in residential cooking and heating and informal industry including brick production. Recent studies (Stockwell et al., 2016) attempted to fill this gap. Such findings must be incorporated into future emission inventory evaluation for further refining regional $PM_{2.5}$ calculations. While the present study did include calculation of both primary and secondary organic matter, as constituents of $PM_{2.5}$, a detailed

study of the sources and fate of total or secondary organic aerosol over the Indian region, is beyond the scope of this work. Direct comparison of spatially averaged model output with satellite products or in-situ measurements typically incorporate significant uncertainty. A broad evaluation was undertaken here, without a match of model output to specific sampling time or satellite overpass time. Thus, some differences would arise from modelled meteorology not faithfully representing actual meteorological conditions during the measurement period. With these caveats, we acknowledge the need for coherent

measurement campaigns to map concentrations of both $PM_{2.5}$ and its chemical constituents over India, to improve model evaluation and future air quality management.

## 4. Simulated $PM_{2.5}$ concentrations by state and sector

### 4.1. Present-day and future $PM_{2.5}$ concentrations at national and state levels

We find that ambient $PM_{2.5}$ pollution is a pan-India problem with a regional character. Figure 6a-g shows the simulated total

ambient $PM_{2.5}$ concentrations for 2015 and in each future scenario (REF, S2, and S3) for 2030 and 2050 to illustrate the different spatial patterns under each scenario. The figure displays mean $PM_{2.5}$ concentration at a grid level, with area-weighted mean values shown in parentheses. Figure 6a shows the simulated annual mean $PM_{2.5}$ concentrations in 2015. It illustrates that the ambient $PM_{2.5}$ concentration has a clear regional distribution with high values in northern India. High $PM_{2.5}$ concentrations in northern India can be attributed both to higher local emissions, especially of organic carbon, and to synoptic transport

patterns leading to confinement of regional emissions of particulate matter and precursor gases in the northern plains (e.g. Sadavarte et al., 2016), borne out in high concentrations of secondary particulate sulphate and dust. In most parts of India values exceed the Indian National Ambient Air-Quality Standard (CPCB, 2009) of 40 µg/m$^3$ for annual mean $PM_{2.5}$, with values as high as 140 µg/m$^3$ in north India. Large regions of north, eastern and western India exhibit high $PM_{2.5}$ concentrations, which are not just limited to specific urban centres or megacities, examined in earlier studies (Jain and Khare, 2008; Guttikunda

et al., 2012; Sharma and Maloo, 2005).

Simulations with the REF scenario emissions (Fig. 6b, c), show significant increases in annual mean $PM_{2.5}$ concentrations all over India, preserving a similar elevated spatial pattern in the north and northeast regions, resulting from significant increases in emissions of primary $PM_{2.5}$ and its precursors from their 2015 values. The REF scenario also results in significant increases, over 2015 levels, in area averaged $PM_{2.5}$ concentrations over India in 2030 (62.3.7%) and 2050 (105.4%) (shown in Fig. 6a,

b, c). The largest future $PM_{2.5}$ concentration values approach 164.1 $\mu g/m^3$ in 2030 and 323.3 $\mu g/m^3$ in 2050 in the REF scenario. Under the S2 scenario, simulated concentrations are projected to improve relative to REF, following similar spatial patterns with the north and northeast regions remaining as the most polluted areas. However, there is no appreciable change in nationally averaged $PM_{2.5}$ concentrations in 2030, while there is even a modest increase in 2050. This implies that energy-use and emission evolution under both current regulation (REF) and that which is promulgated or proposed (S2), are not expected to

yield significant improvements in future air-quality. Under the S3 scenario, a total shift away from traditional biomass technologies and a very large shift (80-85%) to non-fossil electricity generation (S3 scenario) controls the increase in overall $PM_{2.5}$ concentrations and leads to a reduction in spatial variability within India. Under this scenario, the $PM_{2.5}$ concentrations are found to stabilize at 2015 levels without any significant increase in 2030 and 2050 (Fig. 6a, f, g). The mean population-weighted $PM_{2.5}$ concentrations for 2015 and future scenarios for India is shown in Fig. S7 of supplement. The uncertainty

represented by the bars is based on uncertainty in the GBD estimates of ambient $PM_{2.5}$ concentrations. It is estimated by sampling 1,000 draws of a distribution for each grid cell based on the model output mean and standard deviation (GBD MAPS Working Group, 2018).

We further examine what increases or decreases in $PM_{2.5}$ concentrations occur at the state level. India is organized

administratively into 29 states and 7 union territories, therefore, evaluating state-level $PM_{2.5}$ concentrations provides information useful at the regulatory level of state pollution control boards (Air (Prevention and Control of Pollution) Act, 1981). At the state-level, changes in future $PM_{2.5}$ concentrations, from their 2015 levels, were evaluated under the three scenarios (Fig. 7a, b). Simulated $PM_{2.5}$ concentrations from the model are weighted by population for each state. This is calculated by multiplying the concentration in each grid cell (0.1 x 0.1 degree) by the population, summing this quantity for

all grid cells that lie within a state and then dividing by the total population in each state. Under present day emissions of 2015, populations-weighted mean concentrations in most states were above the national $PM_{2.5}$ standard, except for Nagaland, Karnataka, Goa, Manipur, Mizoram, Kerala, Sikkim and Arunachal Pradesh. In 2030, under the REF scenario, significant increases were projected in $PM_{2.5}$ from 2015 levels, in Bihar, Haryana, Jharkhand, Odisha and Uttar Pradesh, while under the S2 scenario, increases were projected in states such as Chhattisgarh, Odisha, and West Bengal. This implies worsening future

air quality in these locations under assumptions of current and promulgated future regulations. However, under the S3 emission scenario which includes control beyond currently promulgated regulations, significant decreases in $PM_{2.5}$ in 2030 were projected with 20 states and six union territories reaching population-weighted mean concentrations below the national ambient air-quality standard, with the largest reductions in Andhra Pradesh, Chhattisgarh, Himachal Pradesh, Odisha. However, 10

states (including Delhi) were projected to continue to have population-weighted mean concentrations above the national $PM_{2.5}$ standard in 2030, even under the lowest emission scenario in this study.

A similar picture was seen in 2050 as well, with very significant increases under the REF scenario in all states, leading to extreme $PM_{2.5}$ concentrations between 100-200 µg/m$^3$, in over ten states (including Bihar, Chhattisgarh, Delhi, Haryana, Jharkhand, Punjab, Uttar Pradesh, West Bengal). Under S2 scenario emissions there was either no appreciable change, or a modest increase in projected $PM_{2.5}$ levels (in states including Andhra Pradesh, Chhattisgarh, Orissa, Telangana and West Bengal). Again, only under S3 scenario emissions, was there a significant reduction in projected future $PM_{2.5}$ levels, with the same 20 states and six union territories falling below the national $PM_{2.5}$ standard; however, the same 10 states (including Delhi) still continue to experience population-weighted mean concentrations higher than the standard.

### 4.2. Simulated source contributions to present-day and future $PM_{2.5}$ concentrations at national and state levels

The simulated change in sectoral contribution to population-weighted $PM_{2.5}$ concentrations is evaluated both at national (Fig. 8) and at the state level (Fig. 9). The figures show the simulated percentage contributions to $PM_{2.5}$ from residential biomass, anthropogenic dust, power plant coal, industry coal, open burning (agricultural), transportation, fired-brick production and distributed diesel sectors. It is cautioned that the sum of contributions from all subsectors does not add up to the simulated ambient concentration from all emission sources. This results from the nonlinearity in the relationship between emissions and ambient concentrations. Nonlinearity is related to atmospheric motion and to atmospheric reactions which are highly non-linear both in space and time, which lead to formation of secondary $PM_{2.5}$ constituents, like sulphate, nitrate and organic carbon. Further, estimation of the fractional contribution from each sector is based on a difference between pairs of simulations, one based on all sources and a sensitivity simulation in which that source sector is removed. Since source-sector based sensitivity simulations were made only for 2015 and 2050 (but not 2030), the figures depict the contribution of the simulated source-sectors in 2015 and that from the three scenarios in 2050. Source contributions have to be interpreted with caution, since they are calculated relative to the total of all sources for a particular year and a particular scenario.

In 2015, among source-sectors, the single largest contributor to ambient $PM_{2.5}$ was residential biomass fuel use for cooking and heating, followed by anthropogenic dust, industrial and power plant coal burning and the open burning of agricultural residues. Emissions from fired-brick production, transportation and distributed diesel (diesel generator sets), also have some contribution to air pollution. It is noteworthy that outdoor air pollution in present day India is dominated by residential biomass fuel use, which is primarily known to contribute to significant burden of disease in India, via household air pollution exposures ((GBD 2016 Risk Factors Collaborators, 2017)). Prior global analyses have also found evidence for the importance of residential biomass fuel use in India (e.g. Verma et al. 2008; 2011; Philip et al., 2014b; Lelieveld et al. 2015; Silva et al., 2016; Lacey et al., 2017). The dominance of residential biomass fuel emissions is an important underlying cause for the regional nature of air pollution in India, because of the widely dispersed and distributed nature of this uncontrolled source. Overall,

Sources related to human activities were responsible for the largest proportion of the present-day population exposure to PM$_{2.5}$ in India. PM$_{2.5}$ concentrations attributable to sources outside India mainly originates from regions to the west of the country so that their contributions to regional background varies considerably by region. Transboundary pollution is highest in the Northwest regions where it contributes about 15% to 30% (>12 ug/m3) and lowest in the southern part of the country where the contributions are less than 15% (4-8 ug/m3). About 60% of India's mean population-weighted PM$_{2.5}$ concentrations arise from anthropogenic source-sectors, with the balance from "other" sources, windblown dust and extra-regional sources. Leading contributors are residential biomass combustion, power plant and industrial coal combustion and anthropogenic dust (including coal fly-ash, fugitive road dust and trash burning). Total dust (wind-blown and anthropogenic) together contributed 39%, while transportation, brick production, and distributed diesel were other contributors to PM$_{2.5}$.

In 2050, future source contributions, are dominated by power plant coal and industrial coal, in both REF and S2 scenarios, followed by residential biomass. In both REF and S2 scenarios (Fig. 2 and Fig. 3) expansion in electricity generation and industry overtakes emissions offsets, leading to 1.5-2 and 1.75-3 times emission increases, respectively, in emissions of PM$_{2.5}$ and its precursor gases, through to 2050. The future expansion projected in power plant and industrial coal use, in both these scenarios, exceeds the growth in biomass fuel use in the residential sector, which follows population increases. Future source contributions to emissions of PM$_{2.5}$ and precursor gas emissions are about 60% from coal burning in electricity generation and industry, with the remainder from biomass energy use in the residential sector, which is directly reflected in source contributions to ambient PM$_{2.5}$. The power plant coal contribution to PM$_{2.5}$ increases in the REF and S2 scenarios, however, it decreases in the S3 scenario, from assumptions of very high penetration (80-85%) of non-fossil electricity generation. The industrial coal contribution to PM$_{2.5}$ concentrations increases above 2015 levels in all future scenarios, reflecting expansion in industry and related "process emissions." This finding suggests that even more stringent measures than those assumed in the scenarios are needed to reduce the influence of industrial coal combustion on ambient pollution levels.

Interestingly, the influence of residential biomass emissions on PM$_{2.5}$ reduces in 2050, even in the REF scenario, from the relative increase in that of industrial coal. In the S2 and S3 scenarios, assumptions of future shift from residential biomass to cleaner LPG/PNG and advanced low-emission gasifier stoves, leads to its decreased contribution to PM$_{2.5}$ concentrations. In the S3 scenario, assumptions of a complete switch away from traditional residential biomass technologies, leads to this sector having the lowest influence on PM$_{2.5}$ concentrations (less than 1.8%). The validity of such assumptions rests upon careful review and effective implementation of national programmes recently launched for expansion of cleaner residential fuels (Pradhan Mantri Ujjwala Yojana, 2016) as well as sustainable adoption of these low emissions approaches. The influence of anthropogenic dust is projected to increase in REF and S2 scenarios while decreasing observed only in the S3 scenario. On the other hand, the influence of total dust is projected to increase in all future scenarios, largely from decreases in the influence of other PM$_{2.5}$ sources. Total and anthropogenic dust concentrations are projected to increase under all scenarios. Dust from anthropogenic activities (anthropogenic dust) is a larger contributor to total dust in REF (47% of total dust, compared with

23% in 2015) and S2 (36% of total dust), while its contributions in S3 (13%) are low. Overall, in S3, total dust (in this scenario dominated by windblown mineral dust) is the largest contributor to ambient $PM_{2.5}$, as a result of the dramatic reductions in emissions projected for all of the other sectors (including anthropogenic dust) in this ambitious scenario. Further examination is needed of the contribution and amelioration of sources in the "other" category, not simulated separately here, which includes trash burning, urban fugitive dust, residential lighting with kerosene and informal industry related to food and agricultural product processing which relies on traditional technologies and biomass fuel.

The $PM_{2.5}$ concentration from transportation sources remains low (<2 µg/m$^3$) under all scenarios but does not decrease in the ambitious scenario. This is related both to the lower magnitude of transportation emissions, relative to other sources, as well as, the relatively coarse model grid (50 km x 67 km). That the transportation contribution decreases in REF but increases in S3 relative to 2015 reflects competing trends from 2015 to 2050 where emissions per vehicle generally decrease but with an increase in vehicle-km. Specifically, passenger-km increase about 4-fold from 2015 to 2050 but with reductions of 15 to 55% in primary $PM_{2.5}$ emissions along with increases in transport-related SO2 (27 to 73%) and NOx (93 to 121%) emissions, depending on the scenario. Further, emissions from transportation may be affected by reductions in emissions from other sectors and non-linear atmospheric chemistry (e.g., reductions in other combustion sources leaving more ammonia available to react with transportation combustion products to form secondary PM). Indeed, evaluation of simulation results indicates that the sensitivity of nitrate to transportation sources in scenario S2 is larger than the nitrate sensitivity in the REF scenario. This suggests that increased available ammonia in S2, resulting from reductions in emissions from other sectors, leads to increased particulate ammonium nitrate formation associated with transportation emissions, relative to the REF scenario. Furthermore, for a number of reasons --because we are estimating sectoral contributions to ambient $PM_{2.5}$ based on the fractional contribution from each sector, because transportation is small relative to the other sectors and because the spatial pattern of the fraction of transport emissions does vary from scenario to scenario --- it is also possible that the decrease in REF, followed by increases in S2 and S3, is an artefact due to increasing fractional contributions from transport relative to other sectors where the decreases are much more dramatic.

Changes in source contributions to $PM_{2.5}$, between 2015 and 2050, are analysed at state level (Fig. 9), wherein patterns similar to those at the national level are seen. Residential biomass fuel use (Fig. 9a) was the dominant source influencing $PM_{2.5}$ in 2015, on both national and state scale. The trade-off between relative decreases in residential biomass, and increases in industrial coal on future $PM_{2.5}$, is seen in the REF, S2 and S3 scenarios, at the state level. In Fig. 9a (residential biomass) note the red-blue-green lines lie below the black dots, while in Fig. 9c (industrial coal), they all lie above the black dots, and in Fig. 9d (power plant coal) only red-blue lines lie above the black dots. Residential biofuel influence reduces in all scenarios in 2050, reaching between 1-2% at the state level, across all states. Anthropogenic dust (Fig. 9b) show decreasing influence while total dust shows increasing influence on $PM_{2.5}$ in the S3 scenario, even at the state level, for reasons discussed above. There is an increase in the influence of industrial coal (Fig. 9c) on $PM_{2.5}$ in all states under all three scenarios, because of expansion,

for the same grid locations, in industrial production and related "process" emissions, e.g. grinding and milling operations in cement industry, despite improved technology efficiencies assumed in the industrial sector. Industrial emission increases are highest in Andhra Pradesh, Jharkhand, Karnataka, Odisha and Tamil Nadu. Further refinement of scenarios must be made to include more stringent industrial emission control technologies. The power plant coal (Fig. 9d) influence increases in the REF and S2 scenarios in all states, however largest increases are seen in Andhra Pradesh, Chhattisgarh, Odisha, West Bengal and Telangana. Under S3 scenario emissions, the power plant coal influence decreases in all states, but has the largest decreases in the same states as above, indicating that the emissions are influenced by high electricity generation in these states, with uniform assumptions made on the shift to non-fossil generation. However, future $PM_{2.5}$ levels are strongly influenced by industrial and power plant coal use, across most states. The influence of open burning (Fig. 9e) appears to change in 2050 under REF and S2 scenarios, not from absolute changes in open burning, but from changes, relative to decreases in the influence of other sources. However, under S3 scenario emissions, in which a complete phase out of open burning is assumed, there are uniform decreases in all states, leaving a negligible influence. The influence of brick production (Fig. 9f) on $PM_{2.5}$ has a negligible increase in the REF scenario at the national level, however, it shows significant increases at the state levels, from 2015 to 2050, in Bihar, Himachal Pradesh, Punjab, Uttar Pradesh and Uttarakhand, the major brick producing states. While the influence of brick production decreases in almost all states under the S3 scenario, it still contributes about 2% in these states through to 2050. The influence of transportation (Fig. 9g) increases significantly under the S3 scenario in a few states like Bihar, Jharkhand, Uttar Pradesh and West Bengal, a likely artefact from the spatial distribution proxy, which uses district level urban population to distribute on-road gasoline emissions. Gasoline vehicles mostly consist of two-, three- and four-wheeler private vehicles in use in urban areas. In the present regional-scale inventory therefore represented using population, pending improved road based proxies for air-quality studies at urban scales.

Overall, sources significantly influencing $PM_{2.5}$ levels include residential biomass in all regions, open burning of agricultural residues in north India, and power plant and industrial coal combustion in eastern and south India. In north India, $PM_{2.5}$ concentrations arise primarily from residential biomass combustion, followed by the open burning of agricultural residues. In contrast, in eastern and south India, while residential biomass combustion is dominant, coal burning in the power and industrial sector is the next important source. Wind-blown dust contributes significantly to $PM_{2.5}$ in north-west India, while anthropogenic dust (largely coal fly-ash) contributes significantly to $PM_{2.5}$ in eastern and south India. Under an ambitious prospective policies scenario, promoting very large shifts away from traditional biomass technologies and coal-based electricity generation, significant reductions in $PM_{2.5}$ levels are achievable in 2030 and 2050. Future air pollution is dominated by industrial process emissions, reflecting larger expansion in industrial, rather than residential energy demand. Potential future contributions of anthropogenic dust are large, while those from transportation and distributed diesel sources are also projected to increase substantially, although small in comparison to other sources.

## 5. Conclusions

This work represents the most comprehensive examination to date of a systematic analysis of source influence, including all sources, on present and future air pollution on a regional scale over India. Elevated annual mean $PM_{2.5}$ concentrations are a pan-India problem, with a regional character, not limited to urban areas or megacities. Under present day emissions, simulations indicate that population-weighted mean concentrations in most states are above the national $PM_{2.5}$ standard. Under present day (2015) emissions, *residential biomass fuel* use for cooking and heating is the largest single sector influencing *outdoor air pollution* across most of India. The dominance of residential biomass fuel emissions is an important underlying cause for the regional nature of air pollution in India, because of the widely dispersed and distributed nature of this uncontrolled source. Agricultural residue burning is the next important source, especially in north-west and north India. This large influence on an annual basis, suggests even larger impacts during the burning periods (typically Apr-May and Oct-Dec). In eastern and peninsular India, the influence of coal burning in thermal power plants and industry follows that of residential biomass combustion. Anthropogenic dust (including coal fly-ash, mineral matter from combustion and urban fugitive dust), brick production and vehicular emissions are also important sources. Overall, the findings suggest a large regional background of $PM_{2.5}$ pollution (from residential biomass, agricultural residue burning and power plant and industrial coal), subjacent to that from local sources (transportation, brick kilns, distributed diesel) in peri-urban areas and megacities.

If no action is taken, population exposures to $PM_{2.5}$ are likely to increase substantially in India by 2050. Evolution of emissions under current regulation (REF) and promulgated or proposed regulation (S2), yields a deterioration in future air-quality future air-quality in 2030 and 2050. Only under the S3 scenario, of ambitious prospective regulation, not yet formulated, promoting a total shift away from traditional biomass technologies and a very large shift (80-85%) to non-fossil electricity generation, is there an overall reduction in $PM_{2.5}$ concentrations below 2015 levels, both in 2030 and 2050, with 20 states and six union territories projected to reach population-weighted mean concentrations below the national ambient air-quality standard. However, even under the most active reductions envisioned, the 2050 population-weighted mean exposure for the S3 scenario, excluding any impact from windblown mineral dust, is estimated to be nearly three times higher than the WHO Air Quality Guideline. Further exploration of air pollution mitigation measures must address the industrial sector, including process emissions, dispersed sources including trash burning and urban fugitive dust, and traditional technologies in residential lighting and informal industry. This study shows that future emission increases in India, if realized, could have important implications for air pollution and climate change on regional and hemispheric scales. Importantly, a government led initiative for detailed emission inventory development at national state and city levels is needed to support air-quality management. Incorporation of detailed Indian emissions, along with their rationalization to other Asian and global inventories, into multi-model studies over the Indian domain would provide insight into atmospheric processes, still lacking in this region.

## Acknowledgements

Partial support for this work was provided by the Health Effects Institute, Boston. KT acknowledges PhD assistantship from the NCAP-COALESE grant of the Ministry of Environment Forests and Climate Change, India. DBM and SC acknowledge support from NASA (#NNX14AP89G), NSF (#AGS-1148951), and the Minnesota Supercomputing Institute. We acknowledge Dr. Sarath Guttikunda, Co-Director, Urbanemissions.info, Goa, India, for datasets of present-day and future emissions from trash burning and urban fugitive dust.

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

**Tables**

**List of Tables**

**Table 1.** Description of source categories and sensitivity simulations

| | Sectors | Source categories | Acronym | Description of sensitivity simulations[a] | |
|---|---|---|---|---|---|
| 1 | Power Plant coal | Thermal power plants | PCOL | Emissions from coal burning in power plants | |
| 2 | Industrial coal | Heavy and Light Industry | ICOL | Emissions from coal burning in heavy and light industries | |
| 3 | Total coal | Thermal power plants, Heavy and Light industry (sum of 1 and 2) | TCOL | Emissions from coal burning in electricity generation, heavy and light industry | |
| 4 | Transportation | Private (2,3,4 wheelers - gasoline), Public (4 wheelers - diesel), Freight (LDDVs[b], HDDVs[c]) and Railways | TRAN | Emissions from on-road and off-road transport including railways | Sensitivity simulations |
| 5 | Distributed Diesel | Agricultural Pumps, Tractors and DG[d] sets | DSDL | Emissions from agricultural pumps, tractors and diesel generator sets | |
| 6 | Residential Biomass | Cooking, Water heating, and Space heating | REBM | Emissions from residential biomass combustion for cooking and heating | |
| 7 | Brick Production | Brick kilns | BRIC | Emissions from brick production | |
| 8 | Open burning | Agricultural residue burning | OBRN | Emissions from agricultural residue burning and forest fires | |
| 9 | Anthropogenic Dust | Mineral matter from combustion and industry, urban fugitive dust | ADST | Emissions of anthropogenic dust. | |
| 10 | Total dust | Windblown mineral dust and anthropogenic dust | TDST | Emissions of dust including windblown mineral dust and from anthropogenic activities. | |
| 11 | Others | Residential lighting (kerosene), Cooking (LPG[e]/Kerosene), Informal industry, Trash burning and Urban fugitive dust | | No sensitivity run was carried out for source categories in this sector except for mineral matter from trash burning and urban fugitive dust (both accounted in ADST). | No sensitivity simulation |
| 12 | Standard | Sum of sectors 1-11, except No 3 | STD | Standard emissions for the year 2015 from all sectors. | Standard simulation |

[a] For each sensitivity simulation, emissions from individual sectors (Nos 1-10) are removed, respectively, from the standard emissions (No 12). Sensitivity simulation results therefore depict the ambient $PM_{2.5}$ concentrations with each emission sector shut off. The differences of the standard and sensitivity simulations were analyzed to produce contributions of the individual sectors to ambient $PM_{2.5}$ concentrations.

10 The "others" sector was not separately addressed in sensitivity simulations. Meteorology was from the year 2012.

[b]LDDVs = Light duty diesel vehicles;  [c]HDDVs = Heavy duty diesel vehicles;  [d]DG= Diesel generator;  [e]LPG = Liquefied petroleum gas

**Table 2.** Description of Future Scenarios

| Source Sectors | REF: Reference Scenario | S2: Aspirational Scenario | S3: Ambitious Scenario |
|---|---|---|---|
| Thermal Power | Low influx of renewable energy with large dominance of sub-critical power plants. | Share of renewable energy (40% by 2030) as targeted in India's NDC with negligible flue gas desulphurization from a slow adoption of recent regulation (MoEFCC, 2015). | 75-80% of non-fossil power generation (Anandarajah and Gambhir 2014; Shukla and Chaturvedi 2012; Level 4, IESS, Niti Aayog, 2015); 80-95% use of flue gas desulphurization. |
| Heavy and Light Industry | Set at present-day efficiency levels (58-75%). | Modest increases in energy efficiency (62-84%) under the Perform Achieve and Trade (PAT) scheme (Level 2, IESS, Niti Aayog, 2015). | Near complete shift to high efficiency (85-100%) industrial technologies (Level 4, IESS, Niti Aayog, 2015). |
| Transport | Present day share of public and private vehicles. | Promulgated growth in public vehicle share (25-30%) (NTDPC, 2013; Guttikunda and Mohan, 2014; NITI Aayog, 2015) with slower shifts to BS-VI standards (MoRTH, 2016 ICRA, 2016). | Large shifts to public vehicles (40-60%) (NITI Aayog, 2015), energy efficiency improvements in engine technology (MoP, 2015) and increased share of electric and CNG vehicle share (20-50%) (NITI Aayog, 2015). |
| Brick and Informal Industry | Largely dominated by traditional technologies such as Bull's trench kilns and clamp kilns. | Modest increases in non-fired-brick walling materials (30-45%) (UNDP, 2009; Maithel, personal communication, 2016). | Large share of non-fired brick walling materials (40-70%) and shift towards use of gasifiers in informal industries (65-80%). |
| Residential | Minor shift (~40%) to energy efficient technologies and fuels. | Slow shift (55% in 2030 and 70% in 2050) to energy efficient technologies and fuels (Level 2, IESS, Niti Aayog, 2015). | Large shifts (90% in 2030 and total in 2050) to LPG and electricity for cooking and heating devices (Level 4, IESS, Niti Aayog, 2015), with complete shift to electric and solar lamps for lighting (National Solar Mission 2010). |
| Agricultural | No reduction in agricultural residue burning. | No reduction in agricultural residue burning. | Slow shift (35% phase out by 2030) and complete phase-out (2050) of agricultural residue burning through a switch to mulching practices (Gupta, 2014). |

**Figures**

**List of Figures**

Figure 1. National Emissions of Particulate matter and Precursor Gases for 2015 (Mt/yr). Emissions of NOx are in Mt yr-1 of NO; SO2 in Mt yr-1 of SO2.

Figure 2. Sectoral emission of fine (a) particulate matter, (d) black carbon and (g) organic carbon under the three scenarios, for 2015-2030 (column 1). Difference of higher efficiency/emission control scenarios from reference(S2 & S3) are shown in column 2 (b,e,h) and column 3 (c,f,i). Emissions from ECLIPSE V5a-CLE and GAINS-WEO2016-NPS are shown for comparison.

Figure 3. Sectoral emission of fine (a) SO2, (d) NOx and (g) NMVOCs under the three scenarios, for 2015-2030 (column 1). Difference of higher efficiency/emission control scenarios from reference (REF & S2) are shown in column 2 (b,e,h) and column 3 (c,f,i). Emissions from ECLIPSE V5a-CLE and GAINS-WEO2016-NPS are shown for comparison.

Figure 4. Model Evaluation by (a) comparison of simulated annual mean $PM_{2.5}$ concentrations with in-situ observations (circles = observations) and (b) comparison of modeled annual mean AOD over India with observations from MODIS.

Figure 5. Evaluation of model performance (NMB) in capturing seasonal variation in chemical species concentrations at two sites in India.

Figure 6. Simulated $PM_{2.5}$ concentration a) 2015 b) 2030 REF c) 2050 REF d) 2030 S2 e) 2050 S2 f) 2030 S3 g) 2050 S3. (Values in the parenthesis represent area-weighted average $PM_{2.5}$ concentration for India).

Figure 7. Population-weighted mean ambient $PM_{2.5}$ concentrations by state for (a) 2015 and 2030 (REF, S2 and S3) and (b) 2015 and 2050 (REF, S2 and S3).

Figure 8. Percentage contribution to ambient $PM_{2.5}$ attributable to different sources in 2015 and 2050 all three scenarios.

Figure 9. Percentage contribution of (a) Residential Biomass, (b) Anthropogenic dust, (c) Industrial coal, (d) Power plant coal, (e) Open burning, (f) Brick production, (g) Transportation and (h) Distributed Diesel attributable to ambient $PM_{2.5}$ concentration by state (2015 – 2050).

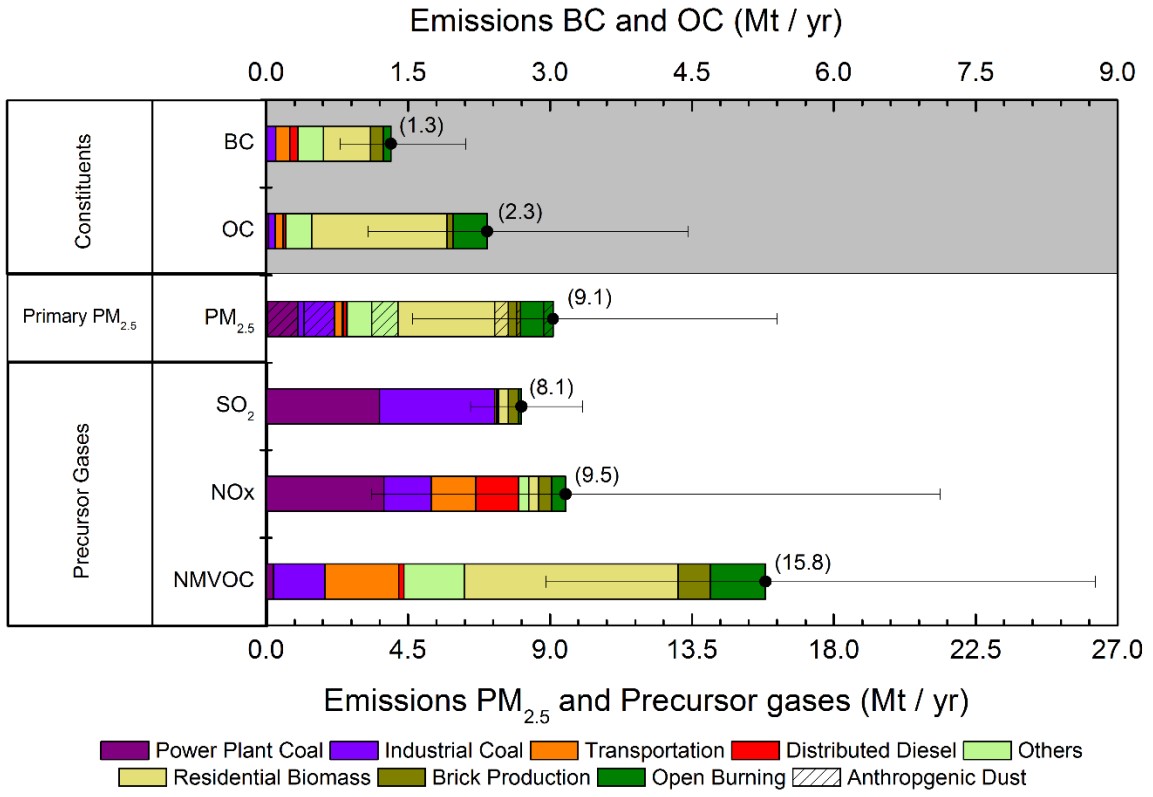

**Figure 1.** National Emissions of Particulate matter and Precursor Gases for 2015 (Mt/yr). Emissions of $NO_x$ are in Mt $yr^{-1}$ of NO; $SO_2$ in Mt $yr^{-1}$ of $SO_2$.

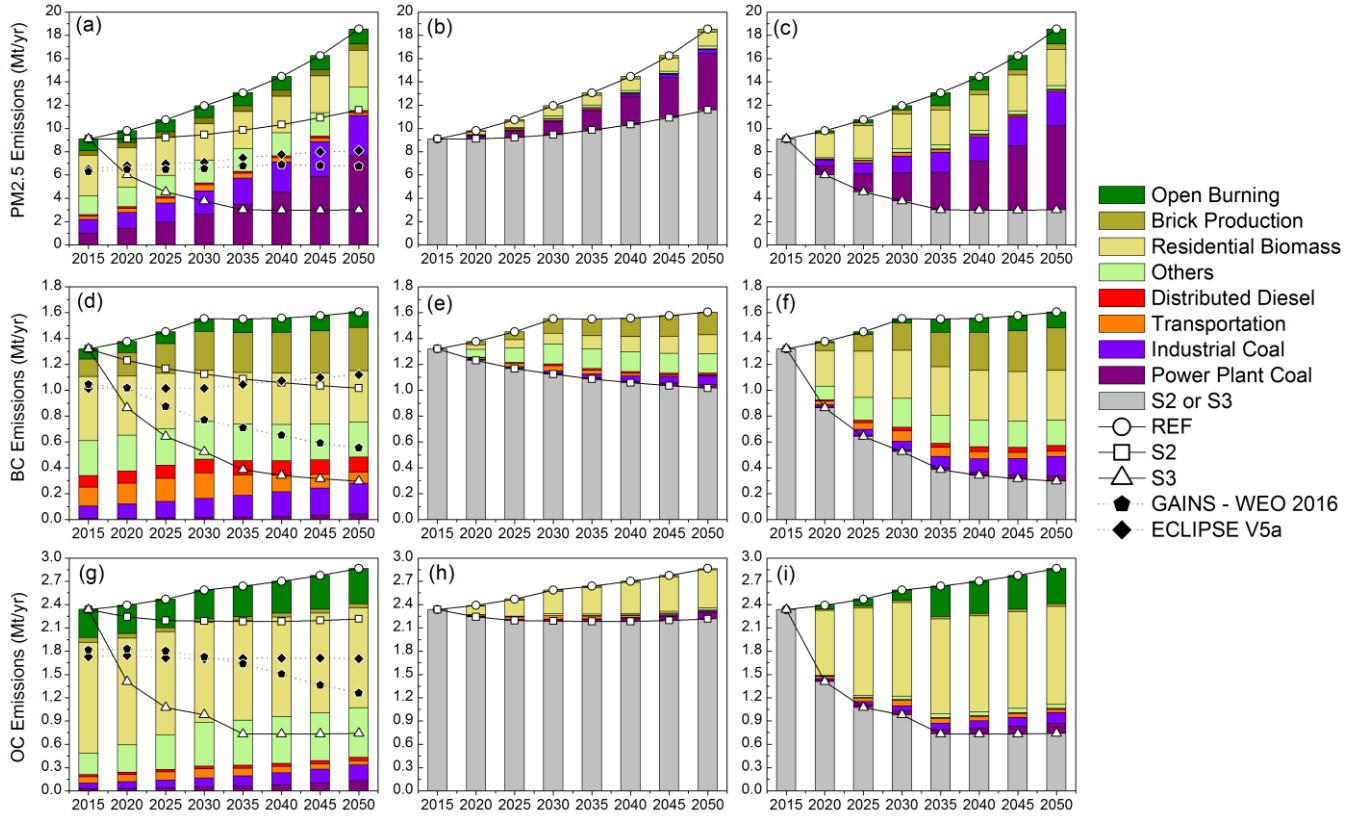

**Figure 2.** Sectoral emission of fine (a) particulate matter, (d) black carbon and (g) organic carbon under the three scenarios, for 2015-2030 (column 1). Difference of higher efficiency/emission control scenarios from reference(S2 & S3) are shown in column 2 (b,e,h) and column 3 (c,f,i). Emissions from ECLIPSE V5a-CLE and GAINS-WEO2016-NPS are shown for comparison.

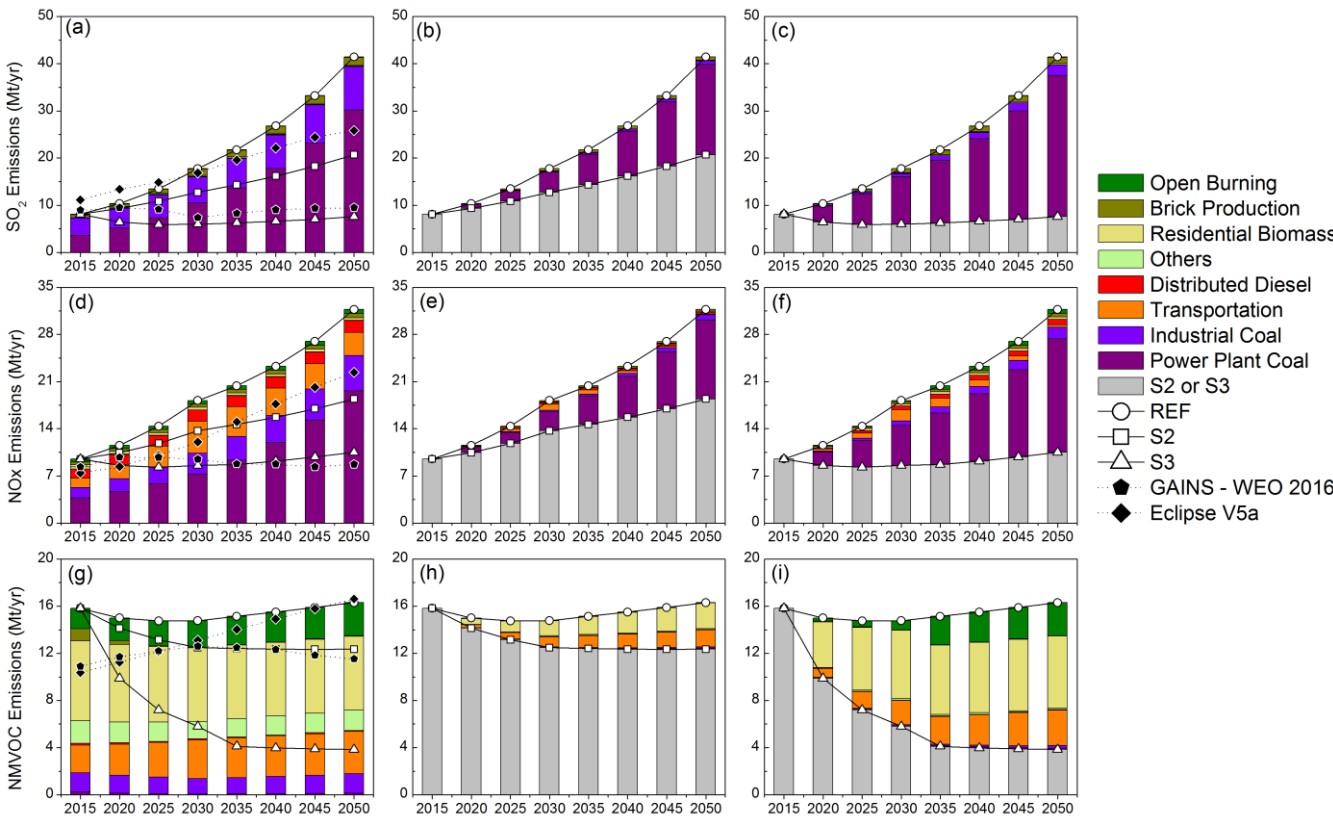

**Figure 3.** Sectoral emission of fine (a) SO$_2$, (d) NO$_x$ and (g) NMVOCs under the three scenarios, for 2015-2030 (column 1). Difference of higher efficiency/emission control scenarios from reference(S2 & S3) are shown in column 2 (b,e,h) and column 3 (c,f,i). Emissions from ECLIPSE V5a-CLE and GAINS-WEO2016-NPS are shown for comparison.

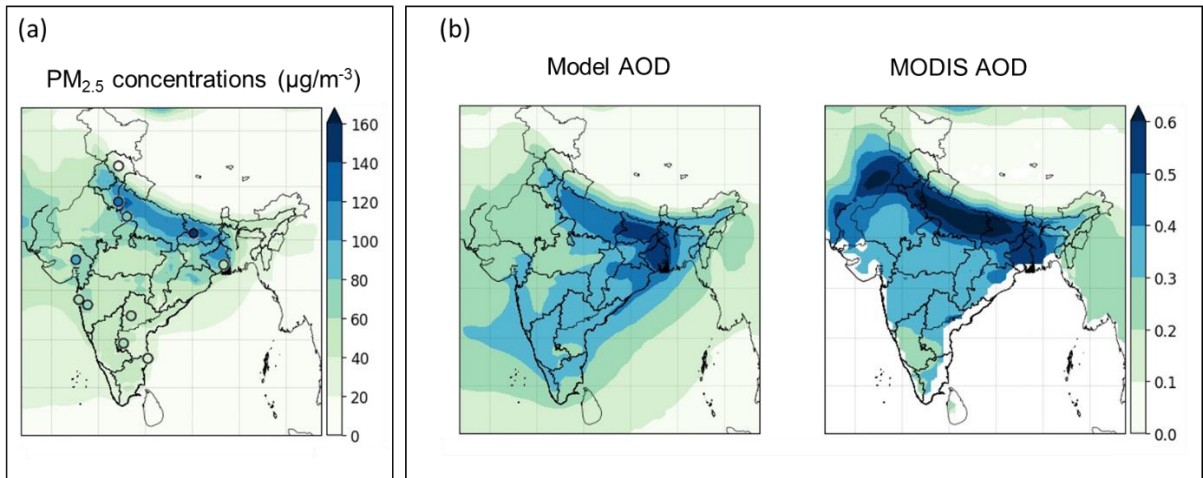

**Figure 4**. Model Evaluation by (a) comparison of simulated annual mean PM$_{2.5}$ concentrations with in-situ observations (circles = observations) and (b) comparison of modeled annual mean AOD over India with observations from MODIS.

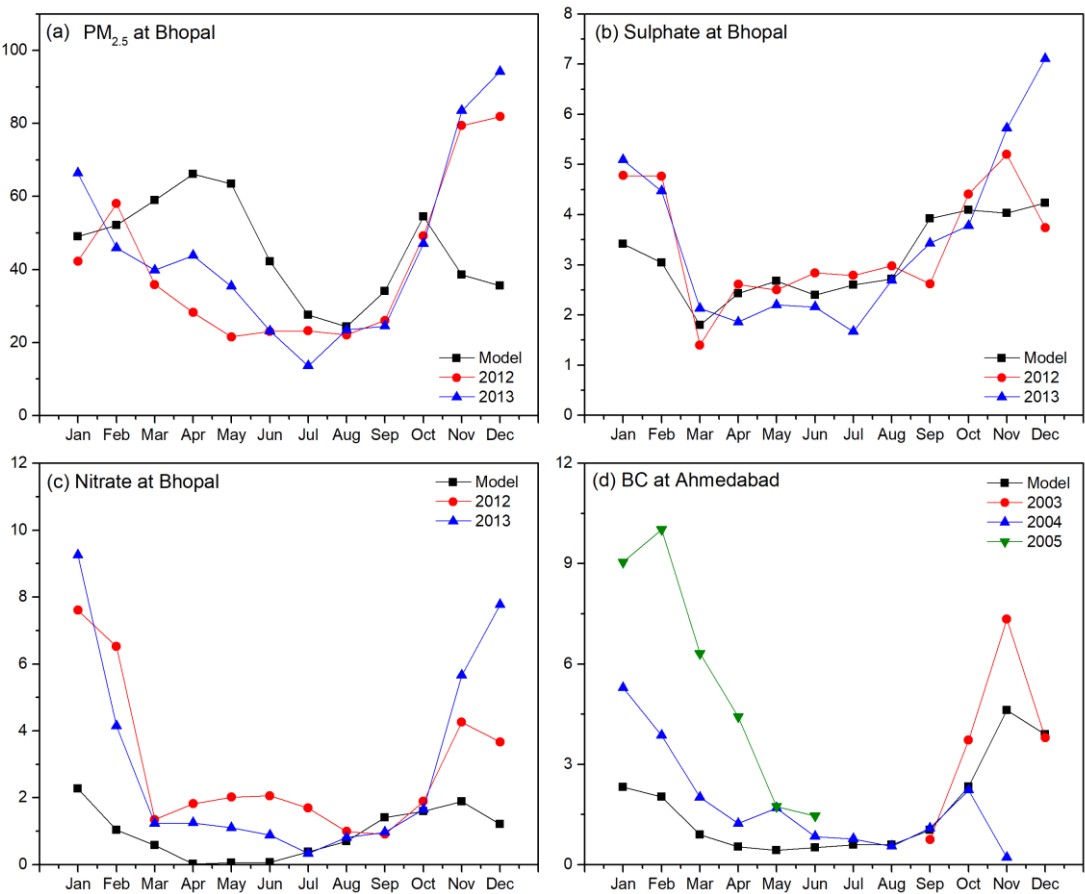

**Figure 5.** Evaluation of model performance (NMB) in capturing seasonal variation in chemical species concentrations at two sites in India

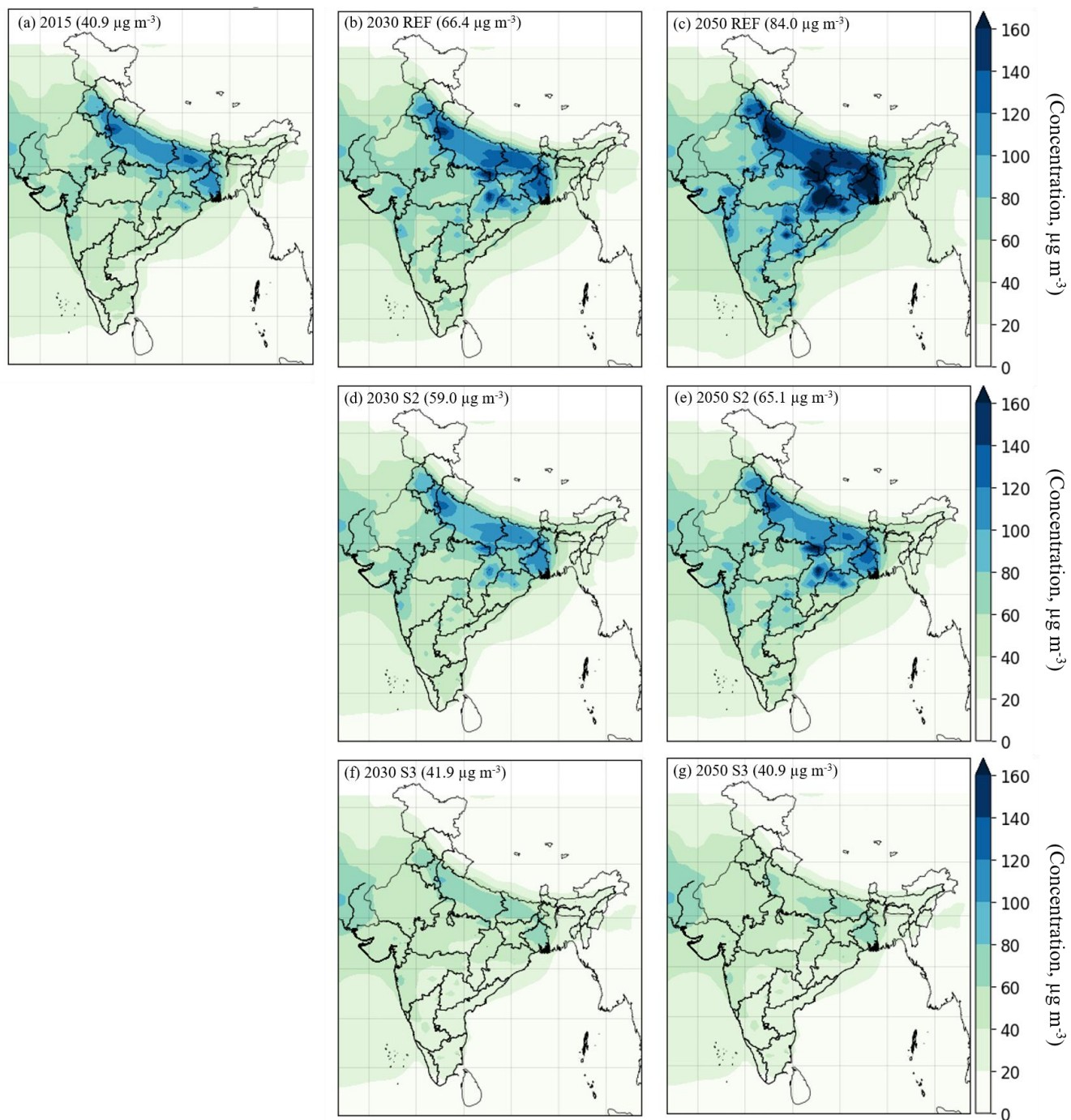

**Figure 6.** Simulated PM$_{2.5}$ concentration a) 2015 b) 2030 REF c) 2050 REF d) 2030 S2 e) 2050 S2 f) 2030 S3 g) 2050 S3. (Values in the parenthesis represent area-weighted average PM$_{2.5}$ concentration for India)

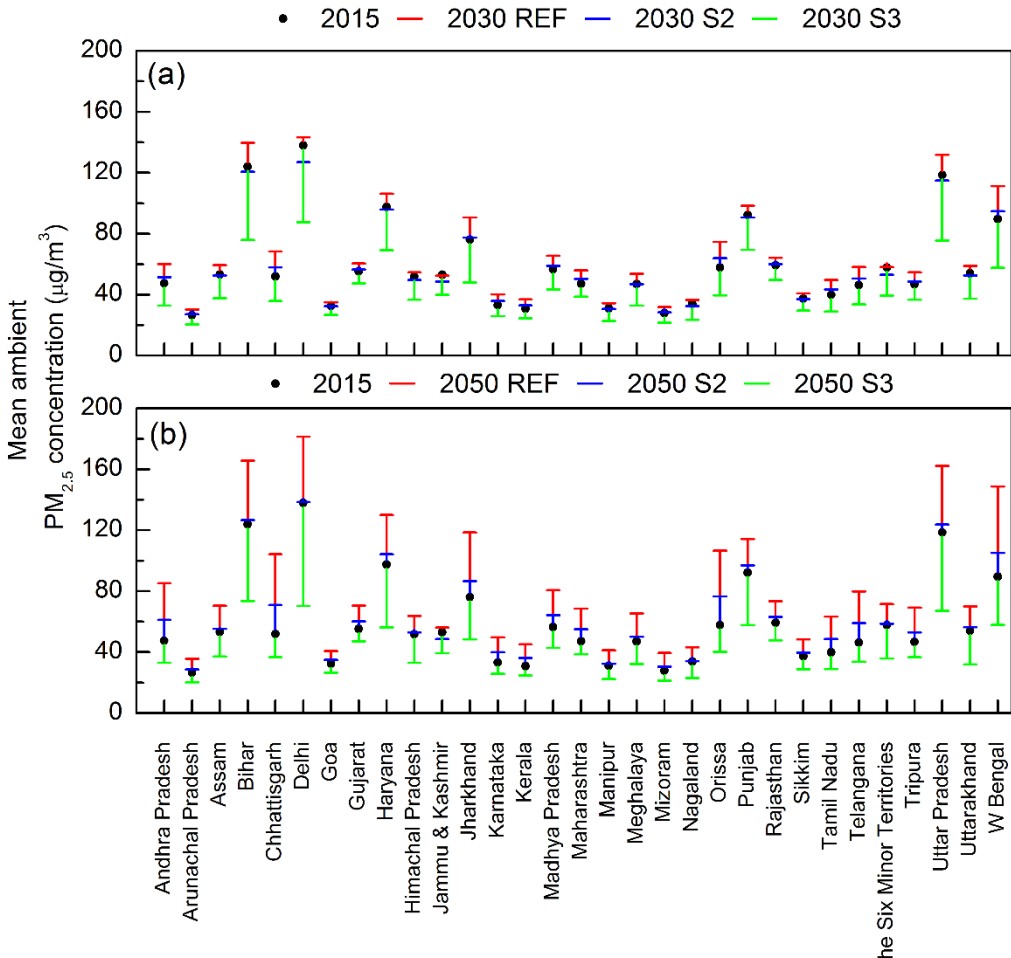

**Figure 7.** Population-weighted mean ambient PM$_{2.5}$ concentrations by state for (a) 2015 and 2030 (REF, S2 and S3) and (b) 2015 and 2050 (REF, S2 and S3)

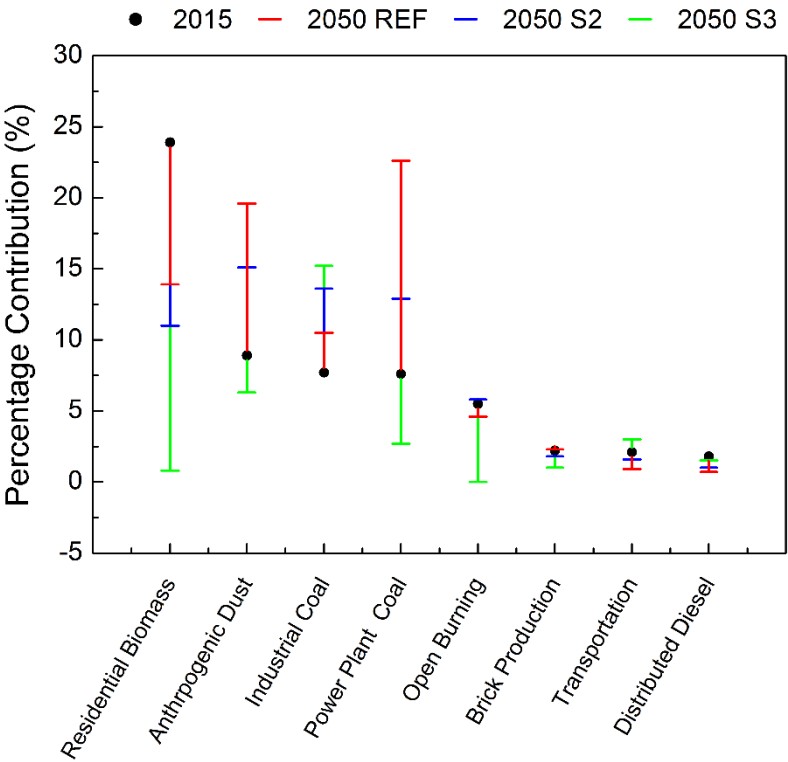

**Figure 8**. Percentage contribution to ambient PM$_{2.5}$ attributable to different sources in 2015 and 2050 all three scenarios.

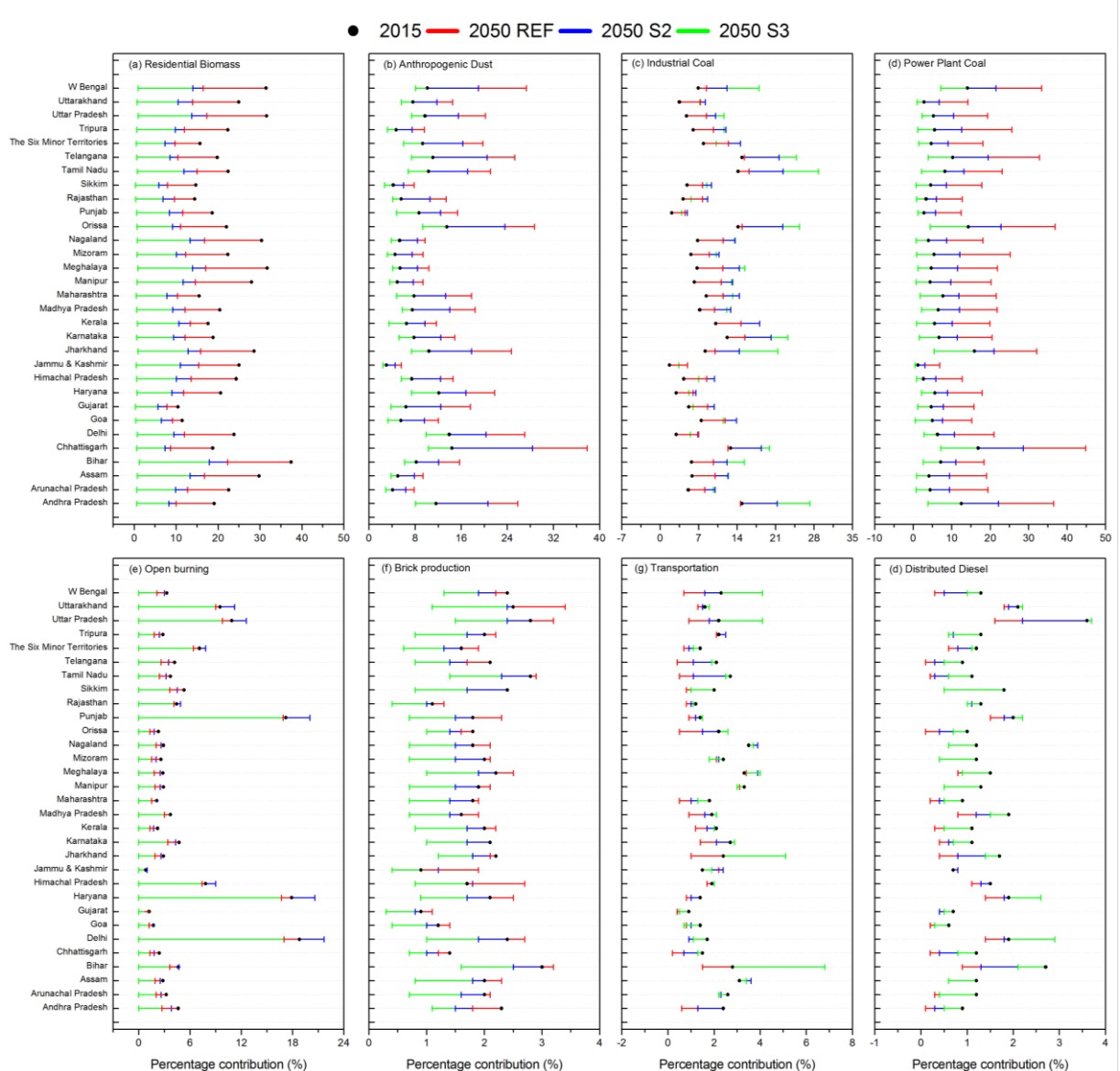

**Figure 9**. Percentage contribution of (a) Residential Biomass, (b) Anthropogenic dust, (c) Industrial coal, (d) Power plant coal, (e) Open burning, (f) Brick production, (g) Transportation and (h) Distributed Diesel attributable to ambient PM$_{2.5}$ concentration by state (2015 – 2050).