# Peer review of "Source influence on emission pathways and ambient PM2.5 pollution over India (2015-2050)"

_Atmospheric Chemistry and Physics, 2017_

## Referee Comment (RC1) · Anonymous Referee #1 · 11 Jan 2018

Review of paper by Chandra Venkataraman et al. titled: **Source influence on emission pathways and ambient PM2.5 pollution over India (2015-2050)**

**General comments**

The work of Venkataraman et al. deals with the investigation of PM sources in India which experiences severe air pollution problems, under current emissions and future emission scenarios which assume cleaner and more energy efficient technologies. This work wants to address two scientific questions strongly related with HTAP, such as the identification of regional PM2.5 pollution levels and their sources and the changes in PM2.5 levels as a result of air pollution and climate change abatement efforts. The paper is overall well written and fits with the purposes of the HTAP special issue; therefore I recommend it for publication after developing the following comments.

**Specific comments**

-page 2 line 5: please provide a reference for the population statistics

-page 2 line 21: in the text you mention that air pollution is a critical issue in particular in certain cities and states of India. It would be interesting to have in the supplementary material a map with the Indian states indicating with markers the most polluted cities.

-page 3 line 10 and page 8 line 27: the HTAP inventory documented by Janssens-Maenhout et al. (2015) is named HTAP_v2, so please correct it.

-page 3 line 27: can you shortly describe the "engineering model approach" on which your emission estimates are based, although documented in other publications. This will help in understanding the source of data for the technology penetrations and air pollution control measures (refer to page 4 line 5).

-page 3 line 8: I guess residential emissions do not only include water and space heating but also all the other domestic activities like cooking. Please correct this sentence.

-page 4 line 20: the authors should clarify why their database does not include emission estimates of CO, NH3 and PM10? Later in the manuscript the authors say that NH3 is indeed taken from MIX. Why was not it possible to calculate them with your methodology? How is the consistency among all pollutants (in terms of activity data, technologies, abatement and spatial distribution) is guaranteed? NH3 is a crucial compound for the formation of secondary PM, so consistency with other SOA precursors is needed. Moreover, you refer to the paper by Li et al., 2017 for the MIX inventory, however, this inventory is only till 2010. How did you obtain emissions for 2015?

-page 5 line 21: The authors mention the "shift to non-fossil generation". Can the authors clarify towards what type of energy source India will move? In addition, as general comment on the

future scenarios, the authors should mention how much realistic/feasible are they. Why Indian emissions cannot increase even at a higher speed compared to 2015 since quite some time is required before future policies to reduce the emissions in India will become effective?

-The authors should compare their scenarios assumptions (including references therein) and results with the recent work by Li et al. (2017).

-page 8 lines 24-43: as supplementary information, it would be interesting to look at some additional emission inventory comparisons for the common years (e.g. 2008 and 2010): e.g. HTAP_v2, REAS, ECLIPSE and your inventory. This can be shown both as total/sector-specific emissions comparison and gridmaps.

-page 11 line 25: why meteorological data are not available beyond 2012?

-page 13 line 14: why do we observe higher concentrations in northern India? Is it only due to the fact that most of the sources are located in that area or are there other reasons?

-page 16 line 13: PM2.5 concentration from road transport seems to be rather low (below 2 ug/m3). Are emissions from re-suspension included?

-page 17 lines 22-24: the authors should clarify why district level urban population is used to distribute on-road gasoline emissions. Transport emissions should be distributed over roads (with different type of weights) and not over population proxies. The authors could provide in a supplementary table the proxies used to grid emissions from different sectors.

-Table1: please clarify what you mean with "emissions of anthropogenic dust removed". If the dust is collected/removed it does not contribute to atmospheric emissions.

-Figure 7 reports PM2.5 concentrations by state, however, it is not clear how this is calculated. Do the authors estimate emissions for each Indian state using statistics of each state and then they evaluate PM2.5 concentrations by state? Please clarify.

-Table S1: it is not clear why NH3 (and possibly also PM10 and CO) emissions by state are not reported here.

-Table S2: it would be good to report a short description in how the uncertainty bands have been calculated using the cited studies.

-Table S4: it would be interesting to know more details about the technologies applied on the private vehicles. The authors could report the share of two/three wheelers and passenger cars as well as the corresponding emission standards (share and emission levels) applied on these vehicles. Is gasoline the most used fuel for private vehicles?

**Technical corrections**

-You should use in the text and in the graphs the "Mt" units instead of "MT"

-page 1 line 30: please rephrase as following: "… and a very large shift (80-85%) to non-fossil electricity generation, an overall reduction in PM2.5 concentrations below 2015 levels was achieved".

-page 2 line 15: please reformulate as following: (particulate matter in a size fraction with diameter smaller than 2.5 μm)

-page 4 line 20: please replace "reside" with "residues".

-page 11 line 21: please correct as following: "mass to organic"

-page 11 line 22: please change to Philip et al. (2014b)

-page 14 line 26: "The simulated change in sectoral contribution to population-weighted PM2.5 concentrations, is evaluated" please remove the "comma"

-page 18 line 15: "The present findings imply that desirable levels of air quality, may not be widespread" please remove the "comma"

-Figure S3 should not be in black and white but with colors.

**References**

Li, C., McLinden, C., Fioletov, V., Krotkov, N., Carn, S., Joiner, J., Streets, D., He, H., Ren, X., Li, Z., and Dickerson, R. R.: India Is Overtaking China as the World's Largest Emitter of Anthropogenic Sulfur Dioxide, Scientific Reports, 7, 14304, 10.1038/s41598-017-14639-8, 2017.

---

## Referee Comment (RC2) · S.nbsp;D. Ghude (Referee) · 21 Feb 2018

This study developed scenarios of sectoral emissions of PM2.5 and its precursors for 2015-2050 and further assessed the impacts of individual source-sectors on PM2.5 pollution through GESO-Chem model simulations over India. Based on model simulations authors have shown that under the present day emissions most states in India exceed NAAQ standard of 40 $\mu$g/m3 (annual mean). Based on emission evaluation under proposed regulations authors have shown further deterioration of air-quality in 2030 and 2050, even in highly ambitious scenario 10 states in India will not meet the current NAAQ standard in 2050. Overall, their finding suggests that residential biomass

burning and agricultural residue burning is the primary largest sector (highly uncertain sector and not validated with the in-situ data) contributing to the large regional background of PM2.5 pollution in India. The paper presents interesting analyses and will be an important resource for the community. However, I have some queries given below and certain key issues need to be addressed for improving the discussion section before it can be accepted for publication. Please find some suggestions below which I hope the authors may find useful for revising the MS for improving the discussion on the issues that affect the uncertainty/certainty of present findings and conclusions.

My major concern is lack of sufficient validation/evaluation of the capability of a well-respected model to simulate chemical species over India, a region with limited publicly available observations. These are very important for meaningful future research too as PM2.5 is a pollutant derived from several precursor emissions with varied sources. Currently the work does not acknowledge such issues and puts too much stock by the model results. Even the model was previously applied to study PM2.5 over India relating satellite AOD to ground-level PM2.5, there has not been a great deal of comparison of model results against observations in previous studies. Global off-line models have large difficulties in simulating chemical species over India (Surenderan et al., 2015, 2016 AE). Therefore it is essential to build confidence in the ability of GEOS-Chem model (since it is finest resolution) to simulate species distributions reasonably well so that it can be used for sensitivity simulations (such as performed for this study) and to understand future air quality projections. Large biases in model may influence the regional PM2.5 fields in the future projections which I believe make it difficult to draw conclusions that are of scientific value. The authors should clearly address this point by comparing the model with the observed PM2.5 for greater understanding of model biases and recognition of areas needing improvement.

As a part of evaluation work for HTAP-II PM2.5 and BC data (mostly from the published literature (not necessary for the same year)) has been compiled for more than 15 stations in India which can be shared to the author for model validation. Of course,

[Figure]

I cannot categorically state that there is a problem, but I do find in figure 4 & 5 that the model has difficulties in simulating the species distribution. There is always a problem of representativeness when comparing coarse-scale models to point observations and perhaps this could be a problem. I would also suggest to the authors to review how they have compared the simulated PM2.5 (model lowest level??) with in-situ observations and satellite AOD (model field interpolated to satellite overpass time). Are NH3 emissions fixed to 2015 level in BAU, S2 and S3 scenarios? NH3 is important compound for the formation of secondary aerosols and agricultural activity is one of the major sources of NH3 in India, particularly in the rural India where residential bio-fuel and biomass burning is dominant. It is necessary to clarify how authors have treated NH3 in 2015 and further in BAU, S2 and S3 scenarios. Considering projected growth in agricultural sector in India it is believed that NH3 emissions will increase further (Sutton et al., 2017). Therefore, it may have some implication on future PM2.5 levels.

Second concern: It is understandable that due to lack of primary measurements concerning several important emission types (e.g. NMVOCs), the magnitude of these emissions are still poorly constrained in the emission inventories and are yet to be validated using in-situ data or with representative emission factors determined from measurements conducted within India from major sources. However, it is necessary to highlight these existing uncertainties arising from the data limiting factors and which are currently substituted through use of emission factors that may not be representative of emission sources in the South Asian atmospheric environment. 1) The authors should provide a speciated list (even in supplement would do) for the NMVOCs considered in this work. Individual NMVOCs have different PM formation potential and without such information it is not possible for the reader to assess how well this class or precursor has been constrained. 2) The key finding reported by the authors concerns the major contribution due to the emissions from traditional biomass technologies in the residential sector (for cooking and heating), the informal industry sector (for brick production and for food and agricultural produce processes), as well as from agricultural reside burning. (Lines17-20; Page 4 of MS). In this regard, it is necessary to point out several

recent studies conducted in Nepal (see Special issue in ACP on Atmospheric pollution in the Himalayan foothills: The SusKat-ABC international air pollution measurement campaign Editor(s): S. S. Gunthe, E. Weingartner, K. O. Nguyen Thi, and E. Stone) and in particular the following papers: Stockwell et al., 2016 and Sarkar et al., 2017). Stockwell et al conducted rare, field measurements in South Asia of emission factors for up to 80 gases (pollutants, greenhouse gases, and precursors) and black carbon for many previously under-sampled sources that are important in developing countries such as cooking with dung and wood, garbage and crop residue burning, brick kilns, motorcycles, generators and pumps, etc. The authors should discuss this work is some detail and compare the emission factor values for reported sources with values used in their work and shown in Table S7. This is important to gauge how much uncertainty can arise from use of variable emission factors. Secondly the work by Sarkar et al. 2017 provides valuable insights on where current emission inventories need to be improved for better representation of emission source contributions. It provides quantitative information regarding the source contributions of the major NMVOC sources in the Kathmandu Valley. Combining high-resolution in situ NMVOC data and model analyses, it showed that REAS v2.1 overestimates the contribution of residential biofuel use and industries. This is very pertinent to discuss and include in the context of the present work for the following reasons. The use of emission factors from residential biofuel sources for determining ambient source contributions without adequately accounting for the deposition and/ or other loss that can occur for the indoor emissions due to household cooking/heating and their net emission to outdoor environment can lead to gross over estimation of the emissions as an atmospheric source. The results of Sarkar et al., 2017, which is focused on NMVOCs appear to point towards such loss processes being significant and if true, this is likely to be even more important for PM2.5 that has higher deposition tendency than gases. These important aspects need to highlighted and addressed so that future work can benefit from such insights. Are there any similar NMVOC datasets reported from the Indian region? It would be good for the authors to mention these if possible. For many of the biomass burning

sources, it is now recognized that combustion efficiency can be even more important than the fuel composition for the emission factors (Roden et al., 2006; Martinsson et al., 2015). Recent relevant work on open agricultural stubble fire emissions of NMVOC from north-west India (Kumar et al., 2018) which appeared after the present work was already in ACPD, may also be helpful for discussing issues pertaining to the inadequate accounting of all gaseous organic gases and uncertainties concerning emission factors.

Minor issues: Page 10, line 30: 'open burning were derive from the global GEFD-4s database' This statement suggests that the authors have used both GEFD-4s open burning emissions as well their own estimated biomass burning emissions for 2015, BAU, S2 and S3. How different GEFD-4s open burning is from the open burning assessed in the present work? Authors should clearly address this point. Page 13, lines 25-30: I have some reservations about the statement made here because sectorial emission distribution is so diverse in India that some regions may see significant change in air quality even in S2 scenario but not necessarily as a regional mean. I would welcome a figure with summary statistics about PM2.5 concentrations for BAU, S2 and S3 scenario for 2105, 2030 and 2050 (e.g., box-whisker plots mean, median, standard deviation, and P25, P75). Page 14, line 15: The term population weighted mean PM2.5 concentration needs to be defined. Page 14, line 28: open burning (agricultural) again how different it is from the GEFD-4s? Pl. make sure that it is now counted double. Page 17, line 7: Is expansion in industrial process in assumed at the same grid locations in BUA, S2 and S3 scenario? If yes, please mention it categorically.

Roden, C. A.; Bond, T. C.; Conway, S.; Pinel, A. B. O. Emission Factors and Real-Time Optical Properties of Particles Emitted from Traditional Wood Burning Cookstoves. Environ. Sci. Technol. 2006, 40 (21), 6750−6757.

Martinsson, J.; Eriksson, A. C.; Nielsen, I. E.; Malmborg, V. B.; Ahlberg, E.; Andersen, C.; Lindgren, R.; Nystrom, R.; Nordin, E. Z.; Brune, W. H.; et al. Impacts of combustion conditions and photochemical processing on the light absorption of biomass combustion

aerosol. Environ. Sci. Technol. 2015, 49, 14663.

Kumar, V., Chandra, B. P. , *Sinha, V., Large unexplained suite of chemically reactive compounds present in ambient air due to biomass fires, Scientific Reports, 8, 626, 2018.

Sarkar, C., Sinha, V., Sinha, B., Panday, A. K., Rupakheti, M. and Lawrence, M. G., Source apportionment of NMVOCs in the Kathmandu Valley during the SusKat-ABC international field campaign using positive matrix factorization, Atmos. Chem. Phys., 17, 8129-8156, 2017. Surendran, D.E., et al,Quantifying the sectoral contribution of pollution transport from South Asia during summer and winter monsoon seasons in support of HTAP-2 experiment, Atmospheric Environment, (2016).

Surendran, D.E., , Air quality simulation over South Asia using Hemispheric Transport of Air Pollution version-2 (HTAP-v2) emission inventory and Model for Ozone and Related chemical Tracers (MOZART-4), Atmospheric Environment, (2015).

Stockwell, C. E., Christian, T. J., Goetz, J. D., Jayarathne, T., Bhave, P. V., Praveen, P. S., Adhikari, S., Maharjan, R., DeCarlo, P. F., Stone, E. A., Saikawa, E., Blake, D. R., Simpson, I. J., Yokelson, R. J., and Panday, A. K.: Nepal Ambient Monitoring and Source Testing Experiment (NAMaSTE): emissions of trace gases and light-absorbing carbon from wood and dung cooking fires, garbage and crop residue burning, brick kilns, and other sources, Atmos. Chem. Phys., 16, 11043-11081, https://doi.org/10.5194/acp-16-11043-2016, 2016.

Please also note the supplement to this comment:
https://www.atmos-chem-phys-discuss.net/acp-2017-1114/acp-2017-1114-RC2-supplement.pdf

---

## Author Comment (AC1) · 26 Mar 2018

**Paper Number:**

acp-2017-1114

**Title**

Source influence on emission pathways and ambient PM2.5 pollution over India (2015-2050)

**Author list**

Chandra Venkataraman, Michael Brauer, Kushal Tibrewal, Pankaj Sadavarte, Qiao Ma, Aaron Cohen, Sreelekha Chaliyakunnel, Joseph Frostad, Zbigniew Klimont, Randall V. Martin, Dylan B. Millet, Sajeev Philip, Katherine Walker, Shuxiao Wang

We thank the reviewers for their comments, which have been carefully addressed in detail. A point by point response (in red) is provided below, which have been reflected as revisions to the manuscript.

**[NOTE: All page and line numbers in the response, refer to the revised version of the manuscript]**

**Response to Referee Comments 1:**

**General comments**

The work of Venkataraman et al. deals with the investigation of PM sources in India which experiences severe air pollution problems, under current emissions and future emission scenarios which assume cleaner and more energy efficient technologies. This work wants to address two scientific questions strongly related with HTAP, such as the identification of regional PM2.5 pollution levels and their sources and the changes in PM2.5 levels as a result of air pollution and climate change abatement efforts. The paper is overall well written and fits with the purposes of the HTAP special issue; therefore I recommend it for publication after developing the following comments.

**Specific comments**

1)-page 2 line 5: please provide a reference for the population statistics

Page 2 line 7: Required reference cited.

*"India hosts the world's second largest population (UNDP, 2017)"*

*Ref:*

United Nations, Department of Economic and Social Affairs, Population Division (2017). World Population Prospects: The 2017 Revision, Key Findings and Advance Tables. Working Paper No. ESA/P/WP/248.

2)-page 2 line 21: in the text you mention that air pollution is a critical issue in particular in certain cities and states of India. It would be interesting to have in the supplementary material a map with the Indian states indicating with markers the most polluted cities.

Page 2 line 25: Map added in Section 3 of supplementary material and referred to in the manuscript.

*"...India feature in a global list of 100 world cities with the highest PM10 (PM with aerodynamic diameter <10 µm) pollution, with cities like Delhi, Raipur, Gwalior, and Lucknow listed among the world's top 10 polluted cities (WHO, 2014; further details in Figure S6 of supplement)."*

The figure is added in the supplementary material, Figure S6.

[Figure]

| Top 20 polluted cities | |
|---|---|
| 1 Delhi | 11 Jharia |
| 2 Faridabad | 12 Gurugram |
| 3 Bhiwadi | 13 Bareilly |
| 4 Patna | 14 Firozabad |
| 5 Dehradun | 15 Ranchi |
| 6 Varanasi | 16 Jaipur |
| 7 Ghaziabad | 17 Kanpur |
| 8 Muzaffarpur | 18 Lucknow |
| 9 Hapur | 19 Agra |
| 10 Amritsar | 20 Moradabad |

**Fig. S1**. Top 20 polluted cities in India (2016)
(Information taken from Greenpeace, 2018)

*Ref:*

Greenpeace: Airpocalypse II, Assessment of air pollution in Indian cities, 2018.

3)-page 3 line 10 and page 8 line 27: the HTAP inventory documented by Janssens-Maenhout et al. (2015) is named HTAP_v2, so please correct it.

Page 3 line 12 - Corrected in the text.

Page 9 line 5 - Corrected in the text.

4)-page 3 line 27: can you shortly describe the "engineering model approach" on which your emission estimates are based, although documented in other publications. This will help in understanding the source of data for the technology penetrations and air pollution control measures (refer to page 4 line 5).

Page 4 line 1: Description added to the text:

*"An engineering model approach, goes beyond fuel divisions and uses technology parameters for process and emissions control technologies, including technology type, efficiency or specific fuel consumption, and technology-linked emission factors (g of pollutant/ kg of fuel) to estimate emissions."*

5)-page 3 line 8: I guess residential emissions do not only include water and space heating but also all the other domestic activities like cooking. Please correct this sentence.

Clarification:

Yes, the residential sector does contain other activities such as cooking and lighting, but the sentence here refers to the assumption in the seasonality in emissions from certain activities. The seasonality is assumed only for space and water heating activities.

Page 4 line 12: The sentence is reframed to convey this information.

*"Residential sector activities are comprised of cooking and water heating, largely with traditional biomass stoves; lighting, using kerosene lamps; and warming of homes and humans, with biomass fuels. Seasonality is included for water heating and home warming."*

6)-page 4 line 20: the authors should clarify why their database does not include emission estimates of CO, NH3 and PM10? Later in the manuscript the authors say that NH3 is indeed taken from MIX. Why was not it possible to calculate them with your methodology? How is the consistency among all pollutants (in terms of activity data, technologies, abatement and spatial distribution) is guaranteed? NH3 is a crucial compound for the formation of secondary PM, so consistency with other SOA precursors is needed. Moreover, you refer to the paper by Li et al. 2017 for the MIX inventory, however, this inventory is only till 2010. How did you obtain emissions for 2015?

Clarification:

In regard to PM-10, the present inventory does not presently include its calculation, but it can be estimated using the current methodology, in future updates to the inventory.

Page 4, line 28: Discussion added.

*"Emissions of CO are included in the inventory (Pandey et al., 2014; Sadavarte et al., 2014), however, CO was not input to the GEOS-Chem simulations, since it is not central to atmospheric chemistry of secondary PM-2.5 formation on annual time-scales."*

Page 11, line 18: Discussion added.

*"Emissions of NH3 arise primarily from sources like animal husbandry, not addressed in the present inventory. Therefore, they are taken from (Li et al., 2017). Owing to large uncertainties in future emissions, these were held the same in future scenarios, as for 2015. Emission magnitudes*

*of NH3 could affect secondary nitrate, which typically contributes to less than 5% of PM-2.5 mass, thus not influencing overall results in any significant manner."*

7)-page 5 line 21: The authors mention the "shift to non-fossil generation". Can the authors clarify towards what type of energy source India will move? In addition, as general comment on the future scenarios, the authors should mention how much realistic/feasible are they. Why Indian emissions cannot increase even at a higher speed compared to 2015 since quite some time is required before future policies to reduce the emissions in India will become effective?

Page 5, line 28: Discussion added.

*"The S2 scenario assumes shifts to non-fossil generation which would occur under India Nationally Determined Contribution (India's NDC, 2015) in the power sector, consistent with a shift to 40% renewables including solar, wind and hydro power by 2030 (NDC, 2015). The NDC goals of India are suggested to be realistic (CAT, 2017; Ross and Gerholdt, 2017), with achievement of non-fossil share of power generation projected to lie between 38%-48% by 2030, as well as adoption of tighter emission standards for desulphurization and de-NOx technologies in thermal plants (MoEFCC, 2015), at a rate consistent with expected barriers (CSE, 2016). Further, changes assumed in the transport sector reflect promulgated growth in public vehicle share (NTDPC, 2013; Guttikunda and Mohan, 2014; NITI Aayog, 2015) and promulgated regulation (Auto Fuel Policy Vision 2025, 2014, MoRTH, 2016), along with realistic assumptions of implementation lags in adoption of BS VI standards (ICRA 2016). Other assumptions include modest increases in industrial energy efficiency under the perform achieve and trade (PAT) scheme (Level 2, IESS, Niti Aayog, 2015 );"*

*Ref:*

*CAT: Climate Action Tracker - India, [online] Available from: http://climateactiontracker.org/countries/india/2017.html (Accessed 5 March 2018), 2017.*

*Ross, K. and Gerholdt, R.: Achieving India's Ambitious Renewable Energy Goals: A Progress Report, World Resources Institute, [online] Available from: http://www.wri.org/blog/2017/05/achieving-indias-ambitious-renewable-energy-goals-progress-report (Accessed 5 March 2018), 2017.*

8)-The authors should compare their scenarios assumptions (including references therein) and results with the recent work by Li et al. (2017).

*Li, C., McLinden, C., Fioletov, V., Krotkov, N., Carn, S., Joiner, J., Streets, D., He, H., Ren, X., Li, Z., and Dickerson, R. R.: India Is Overtaking China as the World's Largest Emitter of Anthropogenic Sulfur Dioxide, Scientific Reports, 7, 14304, 10.1038/s41598-017-14639-8, 2017.*

Page 10, line 1: Discussion added.

*"Bottom-up estimates of SO2 emissions from our inventory (Pandey et al., 2014; Sadavarte et al., 2014) are consistent with the recent estimates from the satellite based study (Li et al., 2017) from 2005-2016, both showing a steady growth. Present day emissions of SO2 (8.1 Mt yr-1) are at the lower end of the range of 8.5-11.3 Mt yr-1 suggested by Li et al. 2017. Large future increases in*

*SO2 emissions, estimated here in the REF and S2 scenarios are consistent with findings of Li et al. 2017."*

9)-page 8 lines 24-43: as supplementary information, it would be interesting to look at some additional emission inventory comparisons for the common years (e.g. 2008 and 2010): e.g. HTAP_v2, REAS, ECLIPSE and your inventory. This can be shown both as total/sector-specific emissions comparison and grid-maps.

page 9, line 15: Discussion added in Supplementary material and referred to in the manuscript.

*"Emission magnitudes of PM-2.5 and precursors in present inventory are in good agreement with those in ECLIPSE for 2010, however, those of precursor gases are somewhat lower (about 30%) than those in HTAP_v2 (2010) and REAS 2.1 (2008) (Section 2.6 of supplement)"*

The figure and discussion is added in the supplement section 2.6:

[Figure]

**Fig. S4.** Comparisons of national totals of SLCPs with HTAP_v2, REAS2.1 and ECLIPSE for 2008 and 2010.

*The past emissions for 2008 and 2010 are compared to other datasets ECLIPSE (Stohl et al., 2015), HTAP_v2 (Janssens-Maenhout et al., 2015) and REAS 2.1 (Kurokawa et al., 2013). Overall emissions from ECLIPSE were found to be in good agreement with those from our inventory, with the difference in total emissions lying within 20%. However, major differences are found in power generation sector, industry and residential. The differences can be attributed to emissions from extraction processes of fuels, commercial activities, and quantification of process emissions from industries. HTAP agree well with PM and its constituents but is nearly a factor 1.5-2 greater for $NO_x$, NMVOCs and $SO_2$. The differences can be majorly attributed to emissions from extraction process in the power sector and difference in control for $NO_x$ and $SO_2$. Similar to HTAP_v2, REAS 2.1 also agrees well for BC and OC while has 0.7 times lower PM and nearly 1.5 times higher emissions of $NO_x$, NMVOCs and $SO_2$ as compared to our inventory. The differences mostly*

*come from inclusion of agricultural emissions (such as fertilizer application and manure management of livestock), non-combustion emissions primarily from solvent use, paint use, evaporative emissions from vehicles, emissions from fuel extraction processes and emissions released from soil in REAS 2.1. Other causes of difference include use of different emission factors and methodologies for emissions estimates, particularly for the residential biomass combustion and transportation. In other inventories, activity data are primarily taken from energy consumption estimates by International Energy Agency (IEA), where as in our inventory the activity data is calculated using food consumption at the state level and end-use energy for cooking (Habib et al., 2004) and vehicular sales to arrive at on-road vehicular population considering age of the vehicles (Pandey and Venkataraman, 2014).*

*Ref:*

*Habib, G., Venkataraman, C., Shrivastava, M., Banerjee, R., Stehr, J. W. and Dickerson, R. R.: New methodology for estimating biofuel consumption for cooking: Atmospheric emissions of black carbon and sulfur dioxide from India, Global Biogeochem. Cycles, 18(3), 1–11, doi:10.1029/2003GB002157, 2004.*

10)-page 11 line 25: why meteorological data are not available beyond 2012?

Page 12 line 12: Discussion added.

*"South Asia nested meteorological fields were not yet available post-2012 due to a change in the GEOS assimilation system in 2013. Therefore, we conducted standard simulations to test meteorology from the years 2010 to 2012. We chose the year 2012 as our meteorology year, as the simulation results using this year best represented the mean PM2.5 concentration from 2010 to 2012. A three month initialization period was used to remove the effects of initial conditions."*

11)-page 13 line 14: why do we observe higher concentrations in northern India? Is it only due to the fact that most of the sources are located in that area or are there other reasons?

Page 14, line 22: Discussion added.

*"High PM-2.5 concentrations in northern India can be attributed both to higher local emissions, especially of organic carbon, and to synoptic transport patterns leading to confinement of regional emissions of particulate matter and precursor gases in the northern plains (e.g. Sadavarte et al., 2016), borne out in high concentrations of secondary particulate sulphate and dust."*

*Ref: Sadavarte, P., Venkataraman, C., Cherian, R., Patil, N., Madhavan, B. L., Gupta, T., Kulkarni, S., Carmichael, G. R. and Adhikary, B.: Seasonal differences in aerosol abundance and radiative forcing in months of contrasting emissions and rainfall over northern South Asia, Atmos. Environ., 125, 512–523, doi:10.1016/j.atmosenv.2015.10.092, 2016.*

12)-page 16 line 13: PM2.5 concentration from road transport seems to be rather low (below 2 ug/m3). Are emissions from re-suspension included?

Clarification:

Yes, the emissions from re-suspension dust is included in the "Anthropogenic dust" category. The emissions under the Transport category only include the emissions from combustion in vehicles.

13)-page 17 lines 22-24: the authors should clarify why district level urban population is used to distribute on-road gasoline emissions. Transport emissions should be distributed over roads (with different type of weights) and not over population proxies. The authors could provide in a supplementary table the proxies used to grid emissions from different sectors.

Page 4 line 18: Spatial proxy table added in the supplement information, Table S1 and referred to in the manuscript.

*"Spatial proxies used to estimate gridded emissions over India are described in Table S1 of the supplement."*

Page 19, line 12: Discussion added.

*"Gasoline vehicles mostly consist of two-, three- and four-wheeler private vehicles in use in urban areas. In the present regional-scale inventory therefore represented using population, pending improved road based proxies for air-quality studies at urban scales."*

14)-Table1: please clarify what you mean with "emissions of anthropogenic dust removed". If the dust is collected/removed it does not contribute to atmospheric emissions.

Clarification:

It is a typo error, the word "removed" should not be mentioned in the table and has been deleted.

15)-Figure 7 reports PM2.5 concentrations by state, however, it is not clear how this is calculated. Do the authors estimate emissions for each Indian state using statistics of each state and then they evaluate PM2.5 concentrations by state? Please clarify.

Page 15 line 18: Discussion added.

*"Simulated PM2.5 concentrations from the model are weighted by population for each state. This is calculated by multiplying the concentration in each grid cell (0.1 x 0.1 degree) by the population, summing this quantity for all grid cells that lie within a state and then dividing by the total population in each state."*

16)-Table S1: it is not clear why NH3 (and possibly also PM10 and CO) emissions by state are not reported here.

See response to comment 6.

17)-Table S2: it would be good to report a short description in how the uncertainty bands have been calculated using the cited studies.

Page 4, line 33: Description added in supplementary material and referred to in the manuscript.

*"Uncertainties in the activity rates, calculated analytically using methods described more fully in previous publications (Pandey and Venkataraman 2014; Pandey et al. 2014; Sadavarte and Venkataraman, 2014) are shown in Table S3 of the supplement."*

Description added in Section 3 of the supplement information:

*"Uncertainties in the activity rates were calculated analytically, assuming normal distribution for the underlying uncertainties in all input quantities. For each input: (a) the mean and standard deviation calculated from a set of available (three or more) data points; (b) upper and lower*

*bounds assumed based on two data points; or (c) a representative uncertainty assumed from similar data, where only one data-point exists. Uncertainty in the emission factors was estimated from the standard deviation in the set of compiled emission factors of a particular pollutant from a particular fuel technology combination. If the emission factor being used was taken from a single reported source, the reported rating was quantified using the percentage errors cited in IPCC (2006a,b) and EMEP (2009). The measured emission factors with unspecified uncertainties were assigned the highest-known uncertainty for the same pollutant and those from similar technologies. Wherever emission factor measurements for a technology were not available an emission factor from a similar technology was chosen and assigned 100% uncertainty (<5% of the technologies fall under this category, including fluidized bed combustors and sponge-iron kilns). A spreadsheet-based approach was developed for combining uncertainties in activity rates and emission factors. A normal/lognormal distribution was assumed for when standard deviation was less/greater than 30% of the mean. Uncertainty propagation in the product of two variables was followed using the sum-of-quadrature rule, calculated analytically. The upper and lower emission bounds were calculated using the resultant lognormal parameters (geometric mean and geometric standard deviation)."*

Refs:

*EMEP, 2009. EMEP/EEA Air Pollutant Emission Inventory Guidebook. European Environment Agency, Copenhagen.*

*IPCC, 2006a. IPCC Guidelines for National Greenhouse Gas Inventories. In: Energy, vol. 2.*

*IPCC, 2006b. IPCC Guidelines for National Greenhouse Gas Inventories. In: General Guidance and Reporting, vol. 1.*

18)-Table S4: it would be interesting to know more details about the technologies applied on the private vehicles. The authors could report the share of two/three wheelers and passenger cars as well as the corresponding emission standards (share and emission levels) applied on these vehicles. Is gasoline the most used fuel for private vehicles?

Discussion added in the supplementary information, Section S2.3:

*"Emissions from on-road vehicles are based from a previous study (Pandey and Venkataraman, 2014). The detailed list of vehicle category is included in the study (Table 3, Pandey and Venkataraman, 2014). Two-wheelers contribute the most to the fleet of private vehicles with approximately 82% share, followed by passenger cars (15%) and three-wheelers (3%). For present day, all vehicles are assumed to be compliant with BS III standards with 2 wheelers having the highest emission levels for PM2.5 followed by three wheelers (0.5 times lower) and gasoline cars (0.1 times lower). Private gasoline vehicles consisting of two-, three- and four-wheeler vehicles which consume nearly 14.0 MT/yr gasoline, compared to 5 MT/yr of diesel consumed by 4-wheeler diesel cars  (Pandey and Venkataraman, 2014). Future shifts to BS IV and BS VI emission standards lead to reductions in emission levels by 80% and 90% respectively."*

*Ref: Pandey, A. and Venkataraman, C.: Estimating emissions from the Indian transport sector with on-road fleet composition and traffic volume, Atmos. Environ., 98, 123–133, doi:10.1016/j.atmosenv.2014.08.039, 2014.*

**Technical corrections**

-You should use in the text and in the graphs the "Mt" units instead of "MT"

Corrected

-page 1 line 30: please rephrase as following: "… and a very large shift (80-85%) to non-fossil electricity generation, an overall reduction in PM2.5 concentrations below 2015 levels was achieved".

Rephrased

-page 2 line 15: please reformulate as following: (particulate matter in a size fraction with diameter smaller than 2.5 μm)

Page 2 line 17: Rephrased

-page 4 line 20: please replace "reside" with "residues".

Page 4 line 27: Corrected

-page 11 line 21: please correct as following: "mass to organic"

Page 12 line 10: Corrected

-page 11 line 22: please change to Philip et al. (2014b)

Page 12 line 11: Corrected

-page 14 line 26: "The simulated change in sectoral contribution to population-weighted PM2.5 concentrations, is evaluated" please remove the "comma"

Page 16 line 6: Corrected

-page 18 line 15: "The present findings imply that desirable levels of air quality, may not be widespread" please remove the "comma"

Corrected

-Figure S3 should not be in black and white but with colors.

Figure replaced

**Referee Comments 2:**

This study developed scenarios of sectoral emissions of PM2.5 and its precursors for 2015-2050 and further assessed the impacts of individual source-sectors on PM2.5 pollution through GESO-Chem model simulations over India. Based on model simulations authors have shown that under the present day emissions most states in India exceed NAAQ standard of 40 g/m3 (annual mean). Based on emission evaluation under proposed regulations authors have shown further deterioration of air-quality in 2030 and 2050, even in highly ambitious scenario 10 states in India will not meet the current NAAQ standard in 2050. Overall, their finding suggests that residential biomass burning and agricultural residue burning is the primary largest sector (highly uncertain sector and not validated with the in-situ data) contributing to the large regional background of PM2.5 pollution in India. The paper presents interesting analyses and will be an important resource for the community. However, I have some queries given below and certain key issues need to be addressed for improving the discussion section before it can be accepted for publication. Please find some suggestions below which I hope the authors may find useful for revising the MS for improving the discussion on the issues that affect the uncertainty/certainty of present findings and conclusions.

**First Concern:**

My major concern is lack of sufficient validation/evaluation of the capability of a well respected model to simulate chemical species over India, a region with limited publicly available observations. These are very important for meaningful future research too as PM2.5 is a pollutant derived from several precursor emissions with varied sources.  Currently the work does not acknowledge such issues and puts too much stock by the model results. Even the model was previously applied to study PM2.5 over India relating satellite AOD to ground-level PM2.5, there has not been a great deal of comparison of model results against observations in previous studies. Global off-line models have large difficulties in simulating chemical species over India (Surenderan et al., 2015, 2016 AE). Therefore it is essential to build confidence in the ability of GEOS-Chem model (since it is finest resolution) to simulate species distributions reasonably well so that it can be used for sensitivity simulations (such as performed for this study) and to understand future air quality projections. Large biases in model may influence the regional PM2.5 fields in the future projections which I believe make it difficult to draw conclusions that are of scientific value. The authors should clearly address this point by comparing the model with the observed PM2.5 for greater understanding of model biases and recognition of areas needing improvement. As a part of evaluation work for HTAP-II PM2.5 and BC data (mostly from the published literature (not necessary for the same year)) has been compiled for more than 15 stations in India which can be shared to the author for model validation. Of course, I cannot categorically state that there is a problem, but I do find in figure 4 & 5 that the model has difficulties in simulating the species distribution. There is always a problem of representativeness when comparing coarse-scale models to point observations and perhaps this could be a problem. I would also suggest to the authors to review how they have compared the simulated PM2.5 (model lowest level??) with in-situ observations and satellite AOD (model field interpolated to satellite overpass time).

Clarification:

We appreciate the referee's suggestion to further evaluate model predictions, which is definitely needed. However, this is strongly limited by the availability of coherent speciated PM-2.5 datasets over India. Therefore, we feel that, at the end of a long and detailed study, exploiting all available measurements, it would be difficult to do another intercomparison well, without taking care to understand details of earlier observation periods proposed, effects of interannual variability, the inherent problems of comparing spatially averaged model output to in-situ measurements, making a close match of model output with sampling times, etc. Further, with observations coming from years quite different from that of the simulation, an evaluation of this nature might not yield much further insight into model performance. We have added the discussion below, explicitly acknowledging the need for more detailed model evaluation in future.

Page 14, line 10: Discussion added.

"*Direct comparison of spatially averaged model output with satellite products or in-situ measurements typically incorporate significant uncertainty. A broad evaluation was undertaken here, without a match of model output to specific sampling time or satellite overpass time. Thus, some differences would arise from modelled meteorology not faithfully representing actual meteorological conditions during the measurement period. With these caveats, we acknowledge the need for coherent measurement campaigns to map concentrations of both PM2.5 and its chemical constituents over India, to improve model evaluation and future air quality management.*"

Are NH3 emissions fixed to 2015 level in BAU, S2 and S3 scenarios? NH3 is important compound for the formation of secondary aerosols and agricultural activity is one of the major sources of NH3 in India, particularly in the rural India where residential bio-fuel and biomass burning is dominant. It is necessary to clarify how authors have treated NH3 in 2015 and further in BAU, S2 and S3 scenarios. Considering projected growth in agricultural sector in India it is believed that NH3 emissions will increase further (Sutton et al., 2017). Therefore, it may have some implication on future PM2.5 levels.

Page 11, line 18: Discussion added.

"*Emissions of NH3 arise primarily from sources like animal husbandry, not addressed in the present inventory. Therefore, they are taken from (Li et al., 2017). Owing to large uncertainties in future emissions, these were held the same in future scenarios, as for 2015. Emission magnitudes of NH3 could affect secondary nitrate, which typically contributes to less than 5% of PM-2.5 mass, thus not influencing overall results in any significant manner.*"

**Second concern:**

It is understandable that due to lack of primary measurements concerning several important emission types (e.g. NMVOCs), the magnitude of these emissions are still poorly constrained in the emission inventories and are yet to be validated using in-situ data or with representative emission factors determined from measurements conducted within India from major sources. However, it is necessary to highlight these existing uncertainties arising from the data limiting

factors and which are currently substituted through use of emission factors that may not be representative of emission sources in the South Asian atmospheric environment.

1) The authors should provide a speciated list (even in supplement would do) for the NMVOCs considered in this work. Individual NMVOCs have different PM formation potential and without such information it is not possible for the reader to assess how well this class or precursor has been constrained.

Page 11 line 24: Table added in Supplementary material and referred to in the manuscript.

*"Total NMVOC emissions from India were taken from Sarkar et al (2016). The GEOS-Chem model speciation (Table S10, supplementary material), into eight species, was applied for further input to the photochemical module."*

Table added in Section 2 of the supplementary material:

Table S10. Description of GEOS-CHEM NMVOC species

| Species in GEOS-Chem | Description |
| --- | --- |
| ACET | Acetone |
| ALD2 | Acetaldehyde |
| ALK4 | Lumped ≤ C4 Alkanes |
| C2H6 | Ethane |
| C3H8 | Propane |
| CH2O | Formaldehyde |
| MEK | Methyl Ehtyl Ketone |
| PRPE | Lumped ≤ C3 Alkanes |

*Ref:*

*Sarkar, M., Venkataraman, C., Guttikunda, S. and Sadavarte, P.: Indian emissions of technology-linked NMVOCs with chemical speciation: An evaluation of the SAPRC99 mechanism with WRF-CAMx simulations, Atmos. Environ., 134, 70–83, doi:10.1016/j.atmosenv.2016.03.037, 2016.*

2) The key finding reported by the authors concerns the major contribution due to the emissions from traditional biomass technologies in the residential sector (for cooking and heating), the informal industry sector (for brick production and for food and agricultural produce processes), as well as from agricultural reside burning. (Lines17-20; Page 4 of MS).In this regard, it is necessary to point out several recent studies conducted in Nepal (see Special issue in ACP on Atmospheric pollution in the Himalayan foothills: The SusKat-ABC international air pollution measurement campaign Editor(s): S. S. Gunthe, E. Weingartner, K. O. Nguyen Thi, and E. Stone) and in particular the following papers: Stockwell et al., 2016 and Sarkar et al., 2017). Stockwell et al conducted rare, field measurements in South Asia of emission factors for up to 80 gases (pollutants, greenhouse gases, and precursors) and black carbon for many previously under-sampled sources that are important in developing countries such as cooking with dung and wood, garbage and crop residue burning, brick kilns, motorcycles, generators and pumps, etc. The authors should discuss this work is some detail and compare the emission factor values for reported sources with values

used in their work and shown in Table S7. This is important to gauge how much uncertainty can arise from use of variable emission factors. Secondly, the work by Sarkar et al. 2017 provides valuable insights on where current emission inventories need to be improved for better representation of emission source contributions. It provides quantitative information regarding the source contributions of the major NMVOC sources in the Kathmandu Valley. Combining high-resolution in situ NMVOC data and model analyses, it showed that REAS v2.1 overestimates the contribution of residential biofuel use and industries. This is very pertinent to discuss and include in the context of the present work for the following reasons. The use of emission factors from residential biofuel sources for determining ambient source contributions without adequately accounting for the deposition and/ or other loss that can occur for the indoor emissions due to household cooking/heating and their net emission to outdoor environment can lead to gross over estimation of the emissions as an atmospheric source. The results of Sarkar et al., 2017, which is focused on NMVOCs appear to point towards such loss processes being significant and if true, this is likely to be even more important for PM2.5 that has higher deposition tendency than gases. These important aspects need to highlighted and addressed so that future work can benefit from such insights. Are there any similar NMVOC datasets reported from the Indian region? It would be good for the authors to mention these if possible. For many of the biomass burning sources, it is now recognized that combustion efficiency can be even more important than the fuel composition for the emission factors (Roden et al., 2006; Martinsson et al., 2015). Recent relevant work on open agricultural stubble fire emissions of NMVOC from north-west India (Kumar et al., 2018) which appeared after the present work was already in ACPD, may also be helpful for discussing issues pertaining to the inadequate accounting of all gaseous organic gases and uncertainties concerning emission factors.

Clarification:

*As pointed out by the reviewer, one of the key findings of this work, suggests the significance of residential biomass, informal industry sector and agricultural residue burning, to annual PM-2.5 concentrations. However, the reviewer appears to suggest that NMVOC emissions, which influence atmospheric secondary organic aerosol, could govern the present source attribution. Sensitivity simulations, made in the present study, with and without secondary organic aerosol estimation, not reported in the paper but reproduced here (below, Fig. R1), reveal that surface concentrations of SOA were a negligible contributor to those of PM-2.5 in the present simulations, contributing at most 1-2 ug/m-3 of PM-2.5 mass. Therefore, the source attribution reported in this work, is not influenced much by SOA, but rather a combination of primary PM-2.5 (organic matter, black carbon, mineral matter) and secondary sulphate, which is attributed by source. Details of the GEOS-Chem NMVOC speciation scheme have been added.*

*In terms of outdoor penetration of indoor smoke from residential biomass, it has been estimated that for typical ventilation and particle deposition rates encountered in rural kitchens in India, about 80% or more of the emissions would penetrate to ambient air (Venkataraman et al. 2005). Therefore, we believe that the source attribution estimated in this study, would not be unduly governed by residential biomass emissions, and is thus robust.*

*However, we agree that there continue to be significant gaps in our understanding of the contribution of both primary and secondary organic aerosol to ambient fine particulate matter in the Indian region. The following discussion is added:*

[Figure]

*FIGURE R1: Sensitivity simulation of secondary organic aerosol to annual mean ambient PM-2.5 concentrations over India.*

: Discussion added.

*As discussed earlier, NMVOC emissions from India were taken from a recent technology-linked inventory, deployed in WRF-CAMx and evaluated with satellite and in-situ observations (Sarkar et al. 2016). However, uncertainties still remain to be addressed in the calculation of secondary PM-2.5 constituents, especially secondary organic aerosols, whose precursor NMVOC emissions in developing countries, are still uncertain from lack of speciation measurements under combustion conditions (Roden et al., 2006; Martinsson et al., 2015) typically encountered in traditional technologies in residential cooking and heating and informal industry including brick production. Recent studies (Stockwell et al., 2016) attempted to fill this gap. Such findings must be incorporated into future emission inventory evaluation for further refining regional PM-2.5 calculations. While the present study did include calculation of both primary and secondary organic matter, as constituents of PM-2.5, a detailed study of the sources and fate of total or secondary organic aerosol over the Indian region, is beyond the scope of this work.*

*Ref:*

*Martinsson, J., Eriksson, A. C., Nielsen, I. E., Malmborg, V. B., Ahlberg, E., Andersen, C., Lindgren, R., Nyström, R., Nordin, E. Z., Brune, W. H., Svenningsson, B., Swietlicki, E., Boman, C. and Pagels, J. H.: Impacts of Combustion Conditions and Photochemical Processing on the Light Absorption of Biomass Combustion Aerosol, Environ. Sci. Technol., 49(24), 14663–14671, doi:10.1021/acs.est.5b03205, 2015.*

*Roden, C. A., Bond, T. C., Conway, S. and Pinel, A. B. O.: Emission Factors and Real-Time Optical Properties of Particles Emitted from Traditional Wood Burning Cookstoves, Environ. Sci. Technol., 40(21), 6750–6757, doi:10.1021/es052080i, 2006.*

*Stockwell, C. E., Christian, T. J., Goetz, J. D., Jayarathne, T., Bhave, P. V., Praveen, P. S., Adhikari, S., Maharjan, R., DeCarlo, P. F., Stone, E. A., Saikawa, E., Blake, D. R., Simpson, I. J., Yokelson, R. J. and Panday, A. K.: Nepal Ambient Monitoring and Source Testing Experiment (NAMaSTE): emissions of trace gases and light-absorbing carbon from wood and dung cooking fires, garbage and crop residue burning, brick kilns, and other sources, Atmos. Chem. Phys., 16(17), 11043–11081, doi:10.5194/acp-16-11043-2016, 2016.*

*Venkataraman, C., Habib, G., Eiguren-Fernandez, A., Miguel, A. H., Friedlander, S. K.: Residential Biofuels in South Asia: Carbonaceous Aerosol Emissions and Climate Impacts, Science, 307(5714), 1454–1456, doi:10.1126/science.1104359, 2005.*

**Minor issues:**

M1) Page 10, line 30: 'open burning were derive from the global GEFD-4s database' This statement suggests that the authors have used both GEFD-4s open burning emissions as well their own estimated biomass burning emissions for 2015, BAU, S2 and S3. How different GEFD-4s open burning is from the open burning assessed in the present work? Authors should clearly address this point.

Page 11, line 14: Sentence reframed.

*"In addition to the emissions described in section 2.2.2, other emissions such as open burning except agricultural residue burning, which includes forest fires were derived from the global GFED-4s database"*

M2) Page 13, lines 25-30: I have some reservations about the statement made here because sectorial emission distribution is so diverse in India that some regions may see significant change in air quality even in S2 scenario but not necessarily as a regional mean. I would welcome a figure with summary statistics about PM2.5 concentrations for BAU, S2 and S3 scenario for 2105, 2030 and 2050 (e.g., box-whisker plots mean, median, standard deviation, and P25, P75).

Page 15, line 11: Plot added in Supplementary material and referred to in the manuscript.

*"The mean population-weighted PM2.5 concentrations for 2015 and future scenarios for India is shown in Figure S7 of supplement."*

Figure added in supplement, section 3:

[Figure]

Fig S7. Mean population-weighted ambient PM₂.₅ concentrations for 2015 and future scenarios. The bars represent the 95% Confidence Interval for the estimates.

M3) Page 14, line 15: The term population weighted mean PM2.5 concentration needs to be defined.

Page 3 line 20: Definition added.

*"...followed by aggregation to population-weighted concentrations (estimated as the sum of product of concentration and population for each grid divided by the total population) at both national and state levels."*

M4) Page 14, line 28: open burning (agricultural) again how different it is from the GEFD-4s? Pl. make sure that it is now counted double.

See comment M1.

M5) Page 17, line 7: Is expansion in industrial process in assumed at the same grid locations in BUA, S2 and S3 scenario? If yes, please mention it categorically.

Page 18, line 28: Sentence reframed.

*"...because of expansion, for the same grid locations, in industrial production and related "process" emissions..."*